# CSPO: Constraint-Sensitive Policy Optimization for Safe Reinforcement Learning

Ayoub Belouadah [1]   Sylvain Kubler [1]   Yves Le Traon [1]

## Abstract

Safe reinforcement learning (Safe RL) aims to maximize expected return while satisfying safety constraints, typically modeled as Constrained Markov Decision Processes (CMDPs). While primal-dual methods scale well to deep RL, they often suffer from delayed constraint correction, leading to oscillatory behavior and prolonged safety violations. In this paper, we propose *Constraint-Sensitive Policy Optimization (CSPO)*, a first-order primal-dual method that incorporates local constraint sensitivity into policy updates. CSPO augments the primal objective with a constraint-sensitive correction derived from the shortest signed distance to the safety boundary, enabling smarter recovery steps back to safety, compensating for delayed Lagrange multiplier updates, reducing oscillations near the boundary, and preserving the KKT solutions of the original constrained problem. Experiments on navigation and locomotion benchmarks demonstrate that CSPO achieves faster safety recovery and high reward preservation, resulting in higher constrained returns compared to state-of-the-art primal-dual and penalty-based methods.

## 1. Introduction

RL has shown remarkable success in high-dimensional decision-making problems, including robotic locomotion, autonomous driving, and complex game playing (Wang et al., 2022). A major limitation of standard RL is that it optimizes expected return without explicitly considering safety constraints, which is unacceptable in real-world applications where constraint violations can result in physical damage or unsafe behavior. Safe RL seeks to overcome this, typically formalizing the problem using the CMDP framework (Altman, 2021), where an agent must ensure that the expected cumulative cost remains below a prescribed limit, restricting the set of feasible policies.

A major line of work in safe RL addresses CMDP using **second-order trust-region** methods (Achiam et al., 2017; Yang et al., 2020; Milosevic et al., 2025), which approximate the constrained objective via sequential quadratic programs, while providing strong theoretical guarentees, they require costly Fisher matrix inversions. Empirical studies show that simpler first-order methods can perform comparably or even better on continuous control benchmarks (Ray et al., 2019). This has motivated **primal–dual Lagrangian** methods (Chow et al., 2018; Tessler et al., 2019), which jointly update policies and multipliers using first-order gradients, as well as **penalty methods** (Zhang et al., 2022; Liu et al., 2020) that augment the reward objective with a cost penalty. However, both exhibit limitations. First, penalty methods require carefully tuned (often large) penalty factors; otherwise, they yield either overly conservative policies or persistent constraint violations. Second, primal-dual methods suffer from a well-known *dual-lag* effect (Tessler et al., 2019; Paternain et al., 2019), leading to unsafe behavior when multipliers are small and excessive conservatism when they grow large, thus exhibiting oscillatory learning dynamics (Platt & Barr, 1987). Moreover, safety recovery requires accounting for the *local sensitivity and steepness of the constraint function*. Flat regions may require stronger corrective updates to return to feasibility, whereas steep regions call for more cautious steps to avoid overshooting. Yet, existing methods fail to exploit this structure, penalizing constraint violations uniformly regardless of local constraint steepness. As illustrated in Figure 1 ('Primal-dual') ignoring constraint sensitivity near steep boundaries can cause overshooting and oscillatory behavior, resulting in instability and inefficient safety recovery.

In this paper, we introduce **Constraint-Sensitive Policy Optimization (CSPO)**, a first-order safe RL algorithm that retains the simplicity of primal–dual methods while explicitly exploiting local first-order constraint information. CSPO augments the primal update with a constraint correction that is applied only when violations occur. By scaling this correc-

---

[1]SnT, University of Luxembourg. Correspondence to: Ayoub Belouadah <ayoub.belouadah@uni.lu>, Sylvain Kubler <sylvain.kubler@uni.lu>, Yves Le Traon <yves.letraon@uni.lu>.

*Proceedings of the 43$^{rd}$ International Conference on Machine Learning*, Seoul, South Korea. PMLR 306, 2026. Copyright 2026 by the author(s).

tion using the norm of the constraint gradient, CSPO adapts the strength of constraint enforcement to the local constraint sensitivity, reducing delayed constraint correction and enabling faster, smoother returns to feasibility (see the CSPO dynamics in Figure 1). Importantly, CSPO remains equivalent to the original constrained problem in terms of both the feasible set and optimal solutions. In addition, we introduce three complementary metrics to evaluate safety recovery: **Time-To-Safety (TTS)**, measuring how quickly feasibility is restored, **Reward Preservation (RP)**, quantifying reward retention during recovery, and **Violation frequency (VF)** capturing how often safety constraints are violated throughout training. Our contributions are:

- We propose CSPO, a first-order primal–dual safe RL algorithm that incorporates local constraint sensitivity to accelerate recovery from constraint violations.

- We demonstrate that CSPO preserves the feasible set and optimal solutions of the original constrained problem.

- We introduce three safety metrics (TTS, RP, VF) to evaluate safety recovery dynamics.

- Empirically, CSPO demonstrates faster and more stable safety convergence with strong performance on continuous-control benchmarks.

## 2. Background

### 2.1. Markov Decision Processes

We consider a discounted MDP $\mathcal{M} = (\mathcal{S}, \mathcal{A}, r, P, \mu, \gamma)$, where $\mathcal{S}$ is the state space, $\mathcal{A}$ is the action space, $r : \mathcal{S} \times \mathcal{A} \times \mathcal{S} \to \mathbb{R}$ is the reward function, $P(s' \mid s, a)$ is the transition kernel, $\mu$ is the initial state distribution, and $\gamma \in [0, 1)$ is the discount factor. A stationary policy $\pi$ is a distribution over actions given states, $\pi(a \mid s)$, and induces a trajectory $\tau = (s_0, a_0, s_1, a_1, \dots)$ with $s_0 \sim \mu$, $a_t \sim \pi(\cdot \mid s_t)$, $s_{t+1} \sim P(\cdot \mid s_t, a_t)$. The discounted return of $\pi$ is

$$J(\pi) = \mathbb{E}_{\tau \sim \pi}\left[\sum_{t=0}^{\infty} \gamma^t r_t\right]. \quad (1)$$

We denote by $V^{\pi(s)}$, $Q^\pi(s, a)$, and $A^\pi(s, a)$ the value, state-action value, and advantage functions of $\pi$, defined respectively as $V^\pi(s) = \mathbb{E}_{\tau \sim \pi}[\sum_{t=0}^{\infty} \gamma^t r_t \mid s_0 = s]$, $Q^\pi(s, a) = \mathbb{E}_{\tau \sim \pi}[\sum_{t=0}^{\infty} \gamma^t r_t \mid s_0 = s, a_0 = a]$, and $A^\pi(s, a) = Q^\pi(s, a) - V^\pi(s)$.

### 2.2. Constrained Markov Decision Processes (CMDP)

A CMDP (Altman, 2021) augments a MDP with cost functions $\{c_i\}_{i=1}^m$, where each $c_i : \mathcal{S} \times \mathcal{A} \times \mathcal{S} \to \mathbb{R}_+$ maps

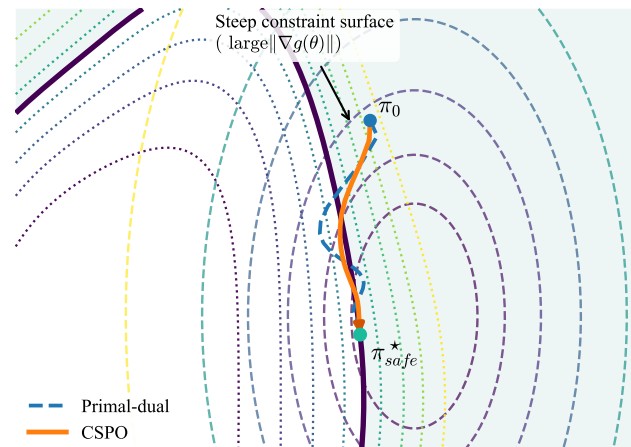

*Figure 1.* Geometric intuition for CSPO on a toy constrained optimization problem. Dashed curves denote objective level sets $f(\theta)$, dotted curves denote constraint level sets $g(\theta)$, the purple curve is the boundary $g(\theta) = 0$, and the shaded region is infeasible. Starting from an infeasible iterate $\pi_0$, standard Lagrangian updates may overshoot and oscillate in highly sensitive regions, whereas CSPO scales the constraint correction by $|\nabla g(\theta)|$, enabling stable recovery to feasibility with reduced reward degradation.

transitions to instantaneous costs. For a stationary policy $\pi$, the discounted cost return for $c_i$ is:

$$J_{c_i}(\pi) = \mathbb{E}_{\tau \sim \pi}\left[\sum_{t=0}^{\infty} \gamma^t c_i(s_t, a_t, s_{t+1})\right]. \quad (2)$$

Given cost limits $d_1, \dots, d_m$, the feasible policy set is

$$\Pi_{\text{safe}} = \{\pi \in \Pi : J_{c_i}(\pi) \le d_i \ \forall i\}. \quad (3)$$

Hence, safe RL aims to find an optimal feasible policy.

$$\pi^\star = \arg\max_{\pi \in \Pi_{\text{safe}}} J(\pi). \quad (4)$$

i.e., maximize return subject to the cost constraints. We similarly define $V_{c_i}^\pi$, $Q_{c_i}^\pi$, and $A_{c_i}^\pi$ as the value, state-action value, and advantage functions associated with the cost function $c_i$ defined respectively as $V_{c_i}^\pi(s) = \mathbb{E}_{\tau \sim \pi}[\sum_{t=0}^{\infty} \gamma^t c_{i_t} \mid s_0 = s]$, $Q_{c_i}^\pi(s, a) = \mathbb{E}_{\tau \sim \pi}[\sum_{t=0}^{\infty} \gamma^t c_{i_t} \mid s_0 = s, a_0 = a]$, and $A_{c_i}^\pi(s, a) = Q_{c_i}^\pi(s, a) - V_{c_i}^\pi(s)$.

### 2.3. Constrained Policy Optimization

Let $d_\pi(s) = (1 - \gamma) \sum_{t=0}^{\infty} \gamma^t \Pr(s_t = s \mid \pi)$ denote the normalized discounted state-visitation distribution of policy $\pi$. For any bounded function $f : \mathcal{S} \times \mathcal{A} \times \mathcal{S} \to \mathbb{R}$, let $J_f(\pi)$ denote the corresponding discounted return, and $A_f^\pi$ its advantage function.

The *performance difference lemma* (Kakade & Langford, 2002) states that, for any two policies $\pi$ and $\pi'$,

$$J_f(\pi') - J_f(\pi) = \frac{1}{1 - \gamma} \mathbb{E}_{s \sim d_{\pi'}, a \sim \pi'}\left[A_f^\pi(s, a)\right]. \quad (5)$$

Applying (5) to the reward $r$ and each cost $c_i$ allows rewriting the safe RL objective (4) as an iterative policy search. For parametrized stochastic policies $\pi_\theta(a \mid s)$ with $\theta \in \mathbb{R}^d$ and a current policy $\pi_{\theta_k}$, the updated policy $\pi_{\theta_{k+1}}$ is obtained by maximizing the reward advantage while satisfying the cost constraints:

$$\pi_{\theta_{k+1}} = \arg\max_{\pi_\theta} \ \mathbb{E}_{s \sim d_{\pi_\theta}, a \sim \pi_\theta} \left[ A_r^{\pi_{\theta_k}}(s, a) \right] \quad (6)$$

$$\text{s.t.} \quad J_{c_i}(\pi_{\theta_k}) + \frac{1}{1-\gamma} \mathbb{E}_{s \sim d_{\pi_\theta}, a \sim \pi_\theta} \left[ A_{c_i}^{\pi_{\theta_k}}(s, a) \right] \ \leq \ d_i,$$

## 3. Constraint-Sensitive Policy Optimization

We consider a differentiable parametric policy class $\pi_\theta : \theta \in \mathbb{R}^d$ with current iterate $\theta_k$. For clarity, we consider a single-constraint case and denote the surrogate cost constraint for policy $\pi_\theta$ as:

$$g(\theta) = J_c(\pi_{\theta_k}) + \frac{1}{1-\gamma} \mathbb{E}_{s \sim d_{\pi_\theta}, a \sim \pi_\theta} \left[ A_c^{\pi_{\theta_k}}(s, a) \right] - d. \quad (7)$$

By definition of the advantage function, this constraint satisfies $g(\theta_k) = J_c(\pi_{\theta_k}) - d$, which corresponds to the constraint violation of the current policy iterate $\pi_{\theta_k}$. We define the feasible parameter set as $\Gamma = \{\theta \in \mathbb{R}^d : g(\theta) \leq 0\}$.

### 3.1. First-Order Geometry of the Constraint

Consider a policy update $\theta_{k+1} = \theta_k + \Delta\theta$. For sufficiently small, trust-region bounded updates and assuming that $g$ is continuously differentiable with locally Lipschitz gradient, the constraint admits the second-order expansion

$$g(\theta_{k+1}) = g(\theta_k) + \nabla_\theta g(\theta_k)^\top \Delta\theta + \mathcal{O}(\|\Delta\theta\|^2). \quad (8)$$

Equivalently, if $\nabla g$ is $L_g$-Lipschitz in a neighborhood of $\theta_k$, then $\left| g(\theta_k + \Delta\theta) - g(\theta_k) - \nabla_\theta g(\theta_k)^\top \Delta\theta \right| \leq \frac{L_g}{2} \|\Delta\theta\|^2$. Under trust-region bounded updates $\|\Delta\theta\| \leq \varepsilon$ for small $\varepsilon > 0$, the approximation error is therefore $\mathcal{O}(\varepsilon^2)$, while the first-order change scales as $\mathcal{O}(\varepsilon)$. Consequently, for small steps, $\mathcal{O}(\|\Delta\theta\|^2)$ is negligible.

During constraint violation, we seek the minimal update to return to the safety boundary. Formulated as:

$$\min_{\Delta\theta} \frac{1}{2} \|\Delta\theta\|^2 \quad \text{s.t.} \quad g(\theta_k) + \nabla_\theta g(\theta_k)^\top \Delta\theta = 0. \quad (9)$$

Solving (9) yields:

$$\Delta\theta^\star = -\frac{g(\theta_k)}{\|\nabla_\theta g(\theta_k)\|^2} \nabla_\theta g(\theta_k), \qquad \|\Delta\theta^\star\| = \frac{g(\theta_k)}{\|\nabla_\theta g(\theta_k)\|} \quad (10)$$

$\nabla_\theta g(\theta_k)$ denotes the gradient of the surrogate constraint with respect to the policy parameters, evaluated at the current iterate $\theta_k$. The length of this update ($\|\Delta\theta^\star\|$) gives us the *shortest signed distance* from $\theta_k$ to the feasible set $\Gamma$.

However, exactly targeting $g(\theta_{k+1}) = 0$ in one update can be overly aggressive under stochastic gradient estimates. Instead, we enforce a *fractional decrease* of the violation, $g(\theta_{k+1}) \leq \sigma \, g(\theta_k), \sigma \in (0, 1]$, which corresponds to targeting the boundary of the *scaled* linearized constraint $g(\theta_k) + \nabla_\theta g(\theta_k)^\top \Delta\theta = \sigma g(\theta_k)$. Solving the same minimal update problem with this target yields the update magnitude:

$$\Delta\theta^\star = -\alpha \frac{g(\theta_k)}{\|\nabla_\theta g(\theta_k)\|^2} \nabla_\theta g(\theta_k), \quad \|\Delta\theta^\star\| = \alpha \frac{g(\theta_k)}{\|\nabla_\theta g(\theta_k)\|} \quad (11)$$

With $\alpha = 1 - \sigma \in [0, 1]$ defined as the fractional reduction factor. See Appendix A for the detailed derivations.

### 3.2. Constraint-Sensitive Objective

The shortest distance formulation in (11) suggests that the magnitude required to reduce a given violation $g(\theta_k)$ depends inversely on the local first-order geometry of the constraint, as captured by $\|\nabla_\theta g(\theta_k)\|$. In particular, large gradient norms correspond to steep constraint surfaces, where overly strong corrections may lead to overshooting and oscillations near the boundary. Conversely, small gradient norms correspond to flat surfaces, where stronger corrective updates are needed for efficient feasibility recovery. As illustrated in Figure 1, accounting for local constraint sensitivity prevents over-correction near steep boundaries while enabling stable recovery toward feasibility aligned with reward level sets. Rather than applying (11) as a direct one-step update, we encode this geometry into a smooth differentiable objective that minimizes the scaled distance to the safety boundary:

$$q_k(\theta) = \frac{\alpha}{2} \, w_k \, [g(\theta)]_+^2, \qquad w_k = \frac{1}{\|\nabla_\theta g(\theta_k)\|^2 + \epsilon} \quad (12)$$

where $\alpha$ is the fractional violation reduction factor from (11), and $\epsilon > 0$ is added for numerical stability. $[x]_+ = \max(x, 0)$ activates the correction only when $g(\theta) > 0$. By construction, the gradient of $q_k$ at $\theta_k$ recovers exactly the parameter update prescribed by (11), turning the geometric recovery step into an objective optimizable with standard gradient-based methods, while vanishing identically at feasible points to preserve equivalence with the original constrained problem. Let $L_R(\theta) = \mathbb{E}_{s \sim d_{\pi_\theta}, a \sim \pi_\theta} \left[ A_r^{\pi_{\theta_k}}(s, a) \right]$, CSPO aims to solve the new equivalent constrained problem:

$$\pi_{\theta_{k+1}} = \arg\min_{\pi_\theta} \ -L_R(\theta) + q_k(\theta) \quad \text{s.t.} \quad g(\theta) \leq 0. \quad (13)$$

**Assumption 3.1.** (Strict feasibility). There exists a constant $\xi > 0$ and a policy $\bar\pi$ such that $g(\bar\theta) \leq -\xi$, i.e., the constraint is satisfied with nonzero margin.

Under Assumption 3.1, we introduce a nonnegative multiplier $\lambda \geq 0$, forming the corresponding Lagrangian of (13):

$$\mathcal{L}_k(\theta, \lambda) = -L_R(\theta) + q_k(\theta) + \lambda\, g(\theta), \qquad \lambda \geq 0. \quad (14)$$

We restrict the dual variable to a compact domain $\Lambda = [0, \lambda_{\max}]$ and consider the associated min-max problem:

$$(\theta_{k+1}, \lambda_{k+1}) \in \arg\min_\theta\ \arg\max_{\lambda \in \Lambda}\ \mathcal{L}_k(\theta, \lambda). \quad (15)$$

We solve (15) iteratively using a primal-dual scheme, performing gradient descent on the policy parameters $\theta$ and projected gradient ascent on the multiplier $\lambda$:

$$\begin{cases} \theta_{k+1} = \theta_k - \eta_\theta \nabla_\theta \mathcal{L}_k(\theta_k, \lambda_k) \\ \lambda_{k+1} = \left[\lambda_k + \eta_\lambda\, g(\theta_k)\right]_+ \end{cases} \quad (16)$$

With $\eta_\theta, \eta_\lambda$ as the primal and dual step sizes. The gradient of the Lagrangian $\mathcal{L}_k(\theta, \lambda)$ takes the form:

$$\nabla_\theta \mathcal{L}_k(\theta, \lambda) = -\nabla_\theta L_R(\theta) + \nabla_\theta q_k(\theta) + \lambda\, \nabla_\theta g(\theta)$$
$$= -\nabla_\theta L_R(\theta) + \left(\lambda + \alpha\, w_k\, [g(\theta)]_+\right) \nabla_\theta g(\theta) \quad (17)$$

Equation (17) shows that CSPO uses an effective multiplier $\lambda_{\text{eff}} = \lambda + \alpha w_k [g(\theta)]_+$, which strengthens constraint correction whenever $g(\theta) \geq 0$. The additional term enables smarter feasibility recovery while allowing $\lambda$ to adapt more smoothly for long-term constraint satisfaction.

**Proposition 3.2.** *Assuming the reward objective $L_R(\theta)$ and the constraint function $g(\theta)$ are continuously differentiable, the original constrained problem (6) and CSPO's augmented problem (13) share the same set of KKT solutions. See Appendix B for proof.*

**Assumption 3.3.** $L_R$ and $g$ are continuously differentiable with $L_R$ being $L_R$-smooth and $g$ being $L_g$-smooth. The gradients and constraint values are uniformly bounded: $\|\nabla g(\theta)\| \leq G_g$, $\|\nabla L_R(\theta)\| \leq G_R$, $|g(\theta)| \leq B_g$. The sensitivity weights satisfy $w_k \leq w_{\max}$, and the dual domain is $\Lambda = [0, \lambda_{\max}]$.

### 3.3. Convergence analysis

Under Assumption 3.3 and appropriate step-size conditions, CSPO's iterates approach an $\varepsilon$-stationary point of (14) at a rate of

$$O\!\left(\frac{L^3 G^2 \lambda_{\max}^2}{\varepsilon^6}\right),$$

where

$$L = L_R + \alpha w_{\max} G_g^2 + (\lambda_{\max} + \alpha w_{\max} B_g)L_g,$$

$$G = G_R + (\lambda_{\max} + \alpha w_{\max} B_g)G_g.$$

This corresponds to an approximate first-order KKT point of the surrogate constrained problem (13), satisfying: (i) $\|\nabla_\theta \mathcal{L}_k(\theta, \lambda)\| \leq \varepsilon$, (ii) $[g(\theta)]_+ \leq \varepsilon$, (iii) $\lambda \geq 0$, (iv) $\lambda[g(\theta)]_+ \leq \varepsilon$, where $\varepsilon$ is controlled by the step-size schedule and problem regularity constants. Appendix C provides the detailed derivation and verifies the required assumptions for the underlying nonconvex-concave minimax formulation.

### 3.4. Extension to Multiple Constraints

CSPO naturally extends to the multi-constraint CMDP setting with $m$ cost constraints. Let $g_i(\theta)$ denote the surrogate constraint corresponding to cost $c_i$. For each constraint, we define:

$$q_k^{(i)}(\theta) = \frac{\alpha_i}{2} w_k^{(i)} [g_i(\theta)]_+^2, \quad w_k^{(i)} = \|\nabla g_i(\theta_k)\|^{-2}.$$

And the Lagrangian becomes

$$\mathcal{L}_k(\theta, \lambda) = -L_R(\theta) + \sum_{i=1}^{m} \left(\lambda_i g_i(\theta) + q_k^{(i)}(\theta)\right), \quad \lambda_i \geq 0.$$

Allowing each constraint to be corrected independently according to its local sensitivity.

## 4. Algorithm

We now describe the practical implementation of CSPO in a deep RL setting. We consider a class of parameterized stochastic policies $\Pi_\theta = \{\pi_\theta(a \mid s) : \theta \in \mathbb{R}^d\}$, represented by neural networks with fixed architecture. Directly minimizing (14) is intractable due to unknown future state distributions and poor sampling efficiency. Following Proximal Policy Optimization (PPO) (Schulman et al., 2017), CSPO uses an analogous clipped surrogate objective for the cost as for the reward. Thus, the practical optimization objective is derived from (14) as:

$$\hat{L}_R(\theta) = \mathbb{E}_t\!\left[\min\!\left(r_t(\theta)A_t^R,\ \text{clip}(r_t(\theta), 1-\epsilon, 1+\epsilon)A_t^R\right)\right] \quad (18)$$
$$\hat{L}_C(\theta) = \mathbb{E}_t\!\left[\max\!\left(r_t(\theta)A_t^C,\ \text{clip}(r_t(\theta), 1-\epsilon, 1+\epsilon)A_t^C\right)\right] \quad (19)$$
$$\hat{g}(\theta) = J_c(\pi_{\theta_k}) - d + \frac{1}{(1-\gamma)}\hat{L}_C(\theta) \quad (20)$$
$$\mathcal{L}_{\text{CSPO}}(\theta) = -\hat{L}_R(\theta) + \frac{\alpha}{2} w\, [\hat{g}(\theta)]_+^2 + \lambda\, \hat{g}(\theta) \quad (21)$$

where the Generalized Advantage Estimator (GAE) (Schulman et al., 2015b) is used to compute the reward and cost advantages $A_t^R$ and $A_t^C$ from trajectories $\{s_t, a_t, r_t, c_t, V_t, V_t^c\}_{t=0}^{\infty}$ collected under $\pi_{\theta_{\text{old}}}$. The importance sampling ratio is $r_t(\theta) = \frac{\pi_\theta(a_t|s_t)}{\pi_{\theta_{\text{old}}}(a_t|s_t)}$. $w = \frac{1}{\|\nabla_\theta \hat{g}(\theta_k)\|^2 + \epsilon}$ is recomputed at the beginning of each policy

---
**Algorithm 1** Constraint-Sensitive Policy Optimization

---
1: Initialize policy network $\pi_{\theta_0}$, value network $V_\psi$, and cost value network $V_\phi^c$
2: Initialize Lagrange multiplier $\lambda_0 \geq 0$
3: **for** episodic iteration $k = 0, 1, 2, \ldots$ **do**
4:     Collect trajectories $\tau \sim \pi_{\theta_k}$
5:     Estimate advantages $A^R, A^C$ using GAE
6:     Update value networks $V_\psi$ and $V_\phi^c$
7:     Compute and detach $w_k = \frac{1}{\|\nabla_\theta \hat{g}(\theta_k)\|^2 + \epsilon}$
8:     **for** $t = 0, 1, \ldots, T-1$ **do**
9:         Compute $\hat{L}_R(\theta)$ in Eq.(18)
10:        Compute $\hat{L}_C(\theta)$ in Eq.(19)
11:        Update policy parameters: $\theta \leftarrow \theta - \eta \nabla_\theta \mathcal{L}(\theta, \lambda_k)$ in Eq.(17)
12:        **if** $\widehat{\mathrm{KL}}(\pi_\theta \| \pi_{\theta_k}) > \delta_{\mathrm{KL}}$ **then**
13:           **break**
14:        **end if**
15:     **end for**
16:     Set $\theta_{k+1} \leftarrow \theta$
17:     Update $\lambda$: $\lambda_{k+1} = \left[\lambda_k + \eta_\lambda (J_c(\pi_{\theta_k}) - d)\right]_+$
18: **end for**

---

update step $k$ using the current policy parameters and is treated as a fixed, detached constant during the subsequent policy updates. For numerical stability, we also incorporate clipping and Exponential Moving Average (EMA) smoothing to $w_k$ (See Appendix G for implementation details). Since $q_k(\theta)$ is continuously differentiable, the surrogate objective (21) can be optimized using standard first-order optimizers (e.g., Adam (Kingma & Ba, 2017)). We present the pseudo-code of CSPO in Algorithm 1. This practical optimization scheme yields the following theoretical properties:

**Proposition 4.1** (Inner-loop stationarity of CSPO). *For a fixed episodic iteration $k$ in Algorithm 1, with $w_k$ and $\lambda_k$ held constant during the inner-loop updates. Let $\{\theta_t\}_{t=0}^{T-1}$ denote the iterates generated by successive policy updates in Algorithm 1 (line 11) applied to the CSPO-augmented objective $\mathcal{L}(\theta, \lambda_k)$. Assuming $\mathcal{L}(\theta, \lambda_k)$ is L-smooth in $\theta$, then the updates satisfy the stationarity rate*

$$\min_{0 \leq t \leq T-1} \left\|\nabla_\theta \mathcal{L}(\theta_t, \lambda_k)\right\|^2 = \mathcal{O}\left(\frac{1}{T}\right) \quad (22)$$

*See Appendix D for proof details.*

**Proposition 4.2** (Local constraint decrease under CSPO updates). *For a fixed episodic iteration $k$ in Algorithm 1, let $\theta_t$ be an inner-loop iterate such that $g(\theta_t) > 0$. Assuming $g$ is continuously differentiable with locally Lipschitz gradient and that $\|\nabla L_R(\theta)\| \leq G_R$ and $\|\nabla g(\theta)\| \leq G_g$ locally. Define $\delta := G_R G_g$. Then for sufficiently small step size $\eta$,*

*one CSPO primal update satisfies*

$$g(\theta_{t+1}) \leq g(\theta_t) - \eta\left(\alpha w\, g(\theta_t)\|\nabla g(\theta_t)\|^2 - \delta\right) + \mathcal{O}(\eta^2) \quad (23)$$

*In particular, if $g(\theta_t) > \frac{\delta}{\alpha w \|\nabla g(\theta_t)\|^2}$, then $g(\theta_{t+1}) < g(\theta_t)$ holds for sufficiently small $\eta$.*

*Remark* 4.3. Under PPO-style clipped surrogates used in (18)-(19) and sufficiently small step size $\eta$, the sufficient condition above simplifies to $g(\theta_t) \gtrsim \frac{\delta}{\alpha}$. See Appendix E for proof.

Proposition 4.2 shows that CSPO guarentees a decrease in constraint violation whenever $g(\theta_t)$ exceeds a threshold proportional to $\frac{1}{\alpha}$. Thus, larger $\alpha$ values activate feasibility-oriented updates at smaller violations, whereas smaller $\alpha$ postpone this activation and permit a wider reward–constraint trade-off before corrective behavior dominates.

**Computational Complexity.** CSPO adds minimal overhead over first-order Lagrangian baselines, requiring one extra surrogate constraint-gradient evaluation per episodic iteration to compute $w_k$, which is detached and kept constant during the inner-loop policy updates.

## 5. Experiments

In this section, we empirically evaluate CSPO. Our experiments are designed to assess the constrained performance, safety recovery behavior, and robustness of CSPO in comparison with state-of-the-art safe RL methods. Specifically, we aim to answer the following questions:

- Does CSPO achieve strong final performance while satisfying safety constraints?

- Does CSPO enable faster recovery to feasibility and mitigating oscillations without sacrificing reward?

- How sensitive is CSPO to different safety parameters?

### 5.1. Experimental Setup

We evaluate CSPO on 9 continuous-control safety tasks (5 locomotion and 4 navigation) from the Safety Gymnasium (Ji et al., 2023) benchmark[1]. The locomotion tasks incentivize forward progress while penalizing excessive velocity, whereas the navigation tasks reward reaching designated goals and impose penalties for entering unsafe regions.

**Baselines.** We compare CSPO against representative baselines from the safe RL literature, including classic Lagrangians (i.e., PPO-Lag (Ray et al., 2019), CPPO-PID (Stooke et al., 2020)), and first-order methods (i.e., FOCOPS

---
[1] https://safety-gymnasium.readthedocs.io/en/latest/

(Zhang et al., 2020), CUP (Yang et al., 2022)), penalty-based methods (i.e., P3O (Zhang et al., 2022), IPO (Liu et al., 2020), and EPO (Gao et al., 2024)) and augmented lagrangians (i.e., APPO (Dai et al., 2023)), and second-order quadratic methods (i.e., CPO (Achiam et al., 2017), PCPO (Yang et al., 2020), C-TRPO (Milosevic et al., 2025)). All methods and experiments have been implemented in the Omnisafe (Ji et al., 2024) library[2] as well as our implementation for CSPO[3]. Detailed hyperparameter settings and training curves are provided in Appendix F.

## 5.2. Overall Performance

We first evaluate the final constrained performance of CSPO. Table 1 reports the Interquartile Mean (IQM) return and cost over the last 100 epochs for all algorithms. Constraint-violating runs (average cost above the threshold) are considered poor performance from the perspective of safe RL. CSPO achieves competitive or superior constrained returns while respecting the cost limits. The gains are most pronounced in navigation tasks (CSPO outperforming all baselines), where precise cost control is required to avoid unsafe regions under sparse rewards. In locomotion environments with velocity-based constraints, CSPO remains competitive with state-of-the-art methods, consistently achieving the best returns while staying close to the safety threshold. Notably, without sensitivity scaling ($w_k = 1$), CSPO reduces to a primal-dual update with a quadratic penalty term (similar in style to APPO) and degenerates to PPO-Lag when $\alpha = 0$. Its consistent improvements over these two, in particular, isolate the benefit of the proposed constraint-sensitive correction, enabling faster feasibility recovery without compromising reward performance across diverse tasks.

## 5.3. Safety metrics evaluation: TTS, RP, VF

To characterize safety behavior beyond average constraint satisfaction, we analyze how frequently algorithms violate constraints and how quickly they recover when violations occur. Let $C_k$ and $J_k$ denote the average episode cost and return at epoch $k$, and let $d$ be the cost limit. We define the empirical constraint violation as $g_k = C_k - d$, where an epoch is safe if $g_k \leq 0$. Three metrics are defined: **Violation frequency (VF):** fraction of training epochs with $g_k > 0$ within a fixed evaluation window (i.e., after 30% of the training budget). **Time to safety (TTS):** measures recovery speed: if a violation begins at epoch $s$ and feasibility is recovered at epoch $e > s$, then TTS $= e - s$. **Reward preservation (RP):** quantifies performance retention during recovery and is defined as RP $= J_e/J_s$. All metrics are averaged over violation episodes and seeds.

Figure 2 visualizes the joint relationship between VF, TTS,

and RP across environments. Across all tasks, CSPO consistently achieves low violation frequency (VF) and fast recovery (TTS), while maintaining competitive reward preservation (RP). The full aggregated results are reported in Table 5. These aggregate trends are consistent with the training dynamics in Figure 3, where CSPO exhibits reduced cost oscillations leading to faster safety recovery compared to Lagrangian baselines.

To better interpret cases where CSPO does not attain the lowest mean TTS, we analyze recovery time versus local constraint sensitivity. We compute $s_t = \|\nabla_\theta g(\theta_t)\|$ at violation states and classify them into flat/steep groups using a quantile split (flat: bottom 30%, steep: top 30%). Table 2 shows that TTS increases in high-sensitivity regions, where CSPO is more conservative to avoid overshooting and preserve reward, while recovery is faster in low-sensitivity regions. This trade-off between fast feasibility recovery in flat regions and reward-preserving recovery in steep regions aligns well with CSPO's final constrained returns in Table 1.

## 5.4. Sensitivity to Safety Parameters

We analyze the sensitivity of CSPO to key safety-related parameters, including the fractional reduction factor $\alpha$ and the cost threshold $d$. Figure 4 ablates $\alpha$ on *PointGoal*, showing it acts as an interpretable safety-recovery *aggressiveness* knob: smaller $\alpha$ yields slower, less intrusive feasibility recovery; while larger $\alpha$ enforces faster constraint satisfaction with temporary reward suppression. For the results reported in Table 1 we used fixed values of $\alpha$ per task family: $\alpha = 0.85$ for locomotion tasks and $\alpha = 0.3$ for navigation tasks. We provide further discussion and practical guidance for choosing $\alpha$ in Section 5.5

We further evaluate the robustness of CSPO to different safety budgets by training under multiple cost thresholds. As shown in Figure 5, CSPO maintains stable behavior across a wide range of cost limits, effectively adapting its performance to the available safety budget while preserving rapid recovery toward feasibility.

Finally, since CSPO is a primal-dual method, we also evaluate its sensitivity to the initialization and learning rate of the Lagrange multiplier. As shown in Appendix I, CSPO remains comparatively robust to these choices, exhibiting reduced sensitivity to the dual hyperparameters.

## 5.5. Practical Choice of $\alpha$

Proposition 4.2 shows that the constraint decrease threshold scales as $\mathcal{O}(1/\alpha)$, directly linking $\alpha$ to the aggressiveness of feasibility recovery. In practice, $\alpha$ can be selected based on the expected severity of constraint violations in the task: when violations are expected to be sudden and large (as in locomotion tasks), larger $\alpha$ promotes faster recovery; when

---

[2] https://github.com/PKU-Alignment/omnisafe
[3] https://github.com/serval-uni-lu/CSPO

*Table 1.* IQM performance and bootstrap 95% CI over 5 seeds on safe RL benchmarks. Cost thresholds $C$ are shown in brackets. We bold the best return among methods that satisfy the constraints. CSPO delivers competitive or superior constrained performance while respecting cost limits, with pronounced gains in navigation tasks.

| Environment | CSPO | APPO | P3O | PPO-Lag | CPO | CUP | FOCOPS | PCPO | CPPOPID | C-TRPO | IPO | EPO |
|---|---|---|---|---|---|---|---|---|---|---|---|---|
| Ant | **3231.6 ± 84.8** | 2961.36 ± 92.37 | 2514.81 ± 81.01 | 3134.17 ± 91.88 | 3055.80 ± 81.74 | 2839.65 ± 113.76 | 2860.88 ± 85.57 | 1728.74 ± 227.2 | 3190.0 ± 69.3 | 3042.64 ± 57.1 | 3123.60 ± 98.79 | 2754.55 ± 113.40 |
| C (25) | 24.5 ± 0.3 | 24.49 ± 1.2 | 23.1 ± 0.5 | 24.3 ± 1.7 | 19.6 ± 1.1 | 26.6 ± 1.8 | 26.7 ± 2.4 | 18.5 ± 1.4 | 25.3 ± 0.4 | 18.0 ± 0.7 | 26.6 ± 0.9 | 13.9 ± 1.1 |
| Humanoid | **6496.85 ± 74.30** | 6092.95 ± 241.32 | 5684.22 ± 87.91 | 6414.56 ± 54.75 | 6213.15 ± 126.53 | 2427.39 ± 504.08 | 5295.88 ± 231.07 | 4706.07 ± 267.57 | 6417.05 ± 54.65 | 5450.52 ± 166.68 | 6270.25 ± 289.6 | 5745.65 ± 110.47 |
| C (25) | 18.0 ± 3.1 | 25.3 ± 0.4 | 22.0 ± 1.1 | 25.49 ± 0.7 | 18.3 ± 2.9 | 18.9 ± 6.1 | 26.7 ± 2.6 | 14.7 ± 1.4 | 26.4 ± 0.6 | 15.9 ± 0.8 | 30.4 ± 4.2 | 13.1 ± 0.5 |
| HalfCheetah | 2007.90 ± 415.95 | 1771.32 ± 417.02 | 1745.84 ± 317.66 | 1841.38 ± 447.36 | 2464.80 ± 366.32 | 1718.19 ± 302.14 | **2870.70 ± 73.78** | 1333.23 ± 39.90 | 2797.52 ± 380.4 | 2748.80 ± 313.62 | 1825.65 ± 326.91 | 2102.11 ± 117.80 |
| C (25) | 15.5 ± 4.1 | 23.1 ± 0.5 | 23.4 ± 0.4 | 24.6 ± 3.6 | 19.7 ± 1.4 | 21.3 ± 3.1 | 20.3 ± 3.5 | 18.3 ± 1.4 | 24.2 ± 0.4 | 15.3 ± 5.2 | 23.8 ± 0.8 | 10.9 ± 2.9 |
| Hopper | **1691.92 ± 70.0** | 1229.86 ± 306.10 | 1255.34 ± 245.60 | 607.20 ± 475.25 | 1639.22 ± 101.83 | 1661.60 ± 11.93 | 1613.64 ± 106.88 | 1112.55 ± 519.68 | 1579.19 ± 351.51 | 1597.48 ± 266.82 | 185.98 ± 9.41 | 1340.49 ± 131.78 |
| C (25) | 7.1 ± 6.1 | 23.6 ± 3.0 | 24.4 ± 1.7 | 22.59 ± 7.1 | 28.1 ± 3.5 | 20.5 ± 9.3 | 19.6 ± 6.2 | 22.5 ± 1.8 | 21.4 ± 2.3 | 23.2 ± 1.6 | 27.1 ± 2.8 | 12.1 ± 5.1 |
| Swimmer | **91.40 ± 56.30** | 74.72 ± 55.90 | 40.94 ± 4.13 | 64.26 ± 19.79 | 44.32 ± 38.58 | 42.39 ± 2.37 | 49.48 ± 17.32 | 32.53 ± 53.41 | 67.12 ± 46.60 | 68.20 ± 54.70 | 45.32 ± 7.40 | 41.85 ± 4.40 |
| C (25) | 4.8 ± 5.1 | 2.6 ± 1.2 | 22.5 ± 3.1 | 30.5 ± 4.6 | 25.0 ± 0.8 | 22.1 ± 2.7 | 30.2 ± 3.7 | 37.0 ± 19.0 | 24.7 ± 1.0 | 16.5 ± 6.9 | 25.6 ± 1.4 | 19.2 ± 0.9 |
| CarButton | **0.70 ± 0.32** | 0.36 ± 0.11 | -0.14 ± 0.20 | 0.52 ± 0.40 | 0.85 ± 0.35 | -0.33 ± 1.30 | 0.63 ± 0.73 | 0.09 ± 0.20 | -1.36 ± 0.23 | 0.65 ± 0.56 | 1.13 ± 0.76 | 0.14 ± 0.60 |
| C (25) | 23.4 ± 0.8 | 24.8 ± 1.8 | 32.9 ± 2.3 | 25.79 ± 2.2 | 33.3 ± 2.8 | 36.2 ± 8.8 | 37.6 ± 5.3 | 35.6 ± 3.6 | 32.7 ± 4.9 | 34.8 ± 1.4 | 34.7 ± 1.0 | 24.2 ± 1.3 |
| CarGoal | **27.11 ± 0.92** | 26.00 ± 1.00 | 21.71 ± 0.92 | 27.70 ± 1.65 | 26.67 ± 0.76 | 20.46 ± 4.02 | 20.78 ± 2.13 | 21.43 ± 0.89 | 6.23 ± 2.16 | 23.07 ± 2.54 | 27.41 ± 0.80 | 25.05 ± 1.16 |
| C (25) | 24.0 ± 0.4 | 24.97 ± 0.8 | 25.7 ± 0.4 | 26.8 ± 2.0 | 27.4 ± 0.6 | 28.2 ± 4.1 | 31.0 ± 2.9 | 29.4 ± 1.5 | 24.6 ± 2.7 | 28.3 ± 0.5 | 27.1 ± 0.6 | 25.2 ± 0.5 |
| PointButton | **10.15 ± 0.20** | 9.01 ± 1.04 | 4.16 ± 1.02 | 9.05 ± 1.65 | 7.02 ± 0.61 | 3.56 ± 1.62 | 5.55 ± 1.52 | 2.61 ± 1.49 | 1.39 ± 1.31 | 6.50 ± 1.55 | 7.92 ± 1.30 | 6.38 ± 1.40 |
| C (25) | 24.4 ± 0.3 | 25.2 ± 0.5 | 26.3 ± 1.7 | 24.1 ± 4.7 | 25.0 ± 0.4 | 30.1 ± 6.0 | 26.8 ± 4.4 | 33.5 ± 3.6 | 24.0 ± 1.7 | 28.9 ± 0.7 | 28.6 ± 0.9 | 24.3 ± 3.8 |
| PointGoal | **23.79 ± 0.75** | 23.12 ± 0.35 | 22.28 ± 0.81 | 21.78 ± 2.38 | 22.58 ± 0.43 | 20.40 ± 3.82 | 15.56 ± 4.36 | 19.82 ± 1.27 | 6.36 ± 3.16 | 19.34 ± 1.03 | 23.35 ± 0.77 | 20.19 ± 1.00 |
| C (25) | 23.7 ± 0.4 | 24.3 ± 0.5 | 24.1 ± 0.6 | 22.0 ± 1.6 | 26.9 ± 0.6 | 25.3 ± 5.5 | 30.8 ± 2.7 | 27.1 ± 0.9 | 25.1 ± 3.7 | 24.8 ± 0.2 | 27.5 ± 0.5 | 24.6 ± 1.4 |

*Table 2.* CSPO recovery metrics conditioned on local constraint sensitivity (Flat: low $\|\nabla g\|$, Steep: high $\|\nabla g\|$).

| Env | Geometry | TTS | RP |
|---|---|---|---|
| Ant | Flat | 3.63 | 1.02 |
| Ant | Steep | 5.25 | 0.99 |
| Humanoid | Flat | 2.33 | 1.00 |
| Humanoid | Steep | 6.17 | 0.99 |
| HalfCheetah | Flat | 4.19 | 1.04 |
| HalfCheetah | Steep | 7.45 | 1.00 |

violations are gradual and bounded, smaller $\alpha$ suffices.

When the violation severity is difficult to characterize a priori, we propose an adaptive schedule motivated by the $\mathcal{O}(1/\alpha)$ scaling of Proposition 4.2, setting $\alpha$ proportionally to the current violation magnitude:

$$\alpha_k = \text{clip}\left(\frac{[J^C(\pi_{\theta_k}) - d]_+}{d + \varepsilon}, 0, 1\right), \qquad (24)$$

so that $\alpha_k$ is small near feasibility and grows under larger violations, providing a practical default that removes the need for task-specific tuning. Normalization by $d$ renders the schedule budget-invariant across different cost thresholds. We compare both the fixed and adaptive schedules empirically in Appendix H.

**5.6. Discussion**

**Originality:** CSPO demonstrates the importance of *constraint sensitivity* in safe policy optimization. Existing primal-dual methods typically apply uniform corrections, ignoring how sensitive the constraint is to parameter updates. This can lead to overshooting and oscillations in high-sensitivity regions, or slow recovery in low-sensitivity regions. In contrast, CSPO scales the constraint correction using the norm of the constraint gradient, enabling stable feasibility recovery using only first-order information. Since the correction activates only under violation,

CSPO preserves equivalence with the original constrained problem.

**Limitations:** While CSPO improves safety recovery dynamics, it relies on accurate cost-gradient estimates, which may become noisy in sparse or discontinuous settings, reducing normalization effectiveness. CSPO also introduces one additional parameter $\alpha$, although bounded and interpretable, its optimal choice remains task-dependent. Designing more robust adaptive or learned sensitivity scaling mechanisms is left for future work.

# 6. Related Work

In this section, we discuss the relevant prior works in the safe RL literature.

**Trust-region methods.** Trust-region approaches such as Constrained Policy Optimization (CPO) (Achiam et al., 2017) extend Trust Region Policy Optimization (TRPO) (Schulman et al., 2015a) by enforcing safety constraints within a local trust region, providing theoretical guarantees under idealized assumptions. More recently, C-TRPO (Milosevic et al., 2025) augments the trust-region formulation with a barrier-based divergence term to ensure that the trust region contains only safe policies. Projection-based variants, including PCPO (Yang et al., 2020), decompose the optimization into a reward maximization step followed by a projection back onto the feasible policy set. In practice, trust-region and projection-based methods require solving quadratic programs using conjugate gradient methods and backtracking line search, resulting in high computational cost and sensitivity to approximation errors.

**Lagrangian and primal-dual methods.** A widely adopted class of Safe RL methods reformulates CMDPs using Lagrangian duality, converting constraints into weighted penalties on expected costs. Representative approaches in-

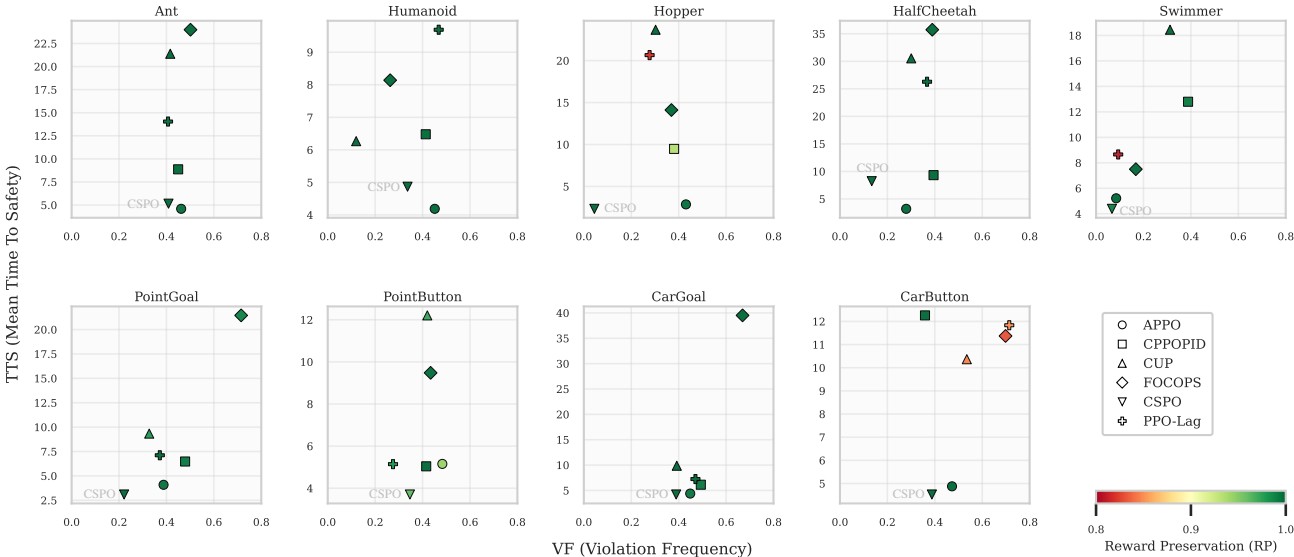

*Figure 2.* Safety recovery behavior across environments. Each point corresponds to one algorithm in a given environment, plotted by violation frequency (VF) and mean time to safety (TTS). Marker color encodes reward preservation (RP), with greener colors indicating better reward retention. Lower VF and lower TTS indicate safer and more stable behavior. Across all tasks, CSPO consistently achieves low VF and low TTS, while maintaining high RP.

clude PPO-Lag and TRPO-Lag variants (Ray et al., 2019), as well as general primal–dual optimization methods (Chow et al., 2018; Paternain et al., 2019; Tessler et al., 2019). While simple and scalable, these methods often suffer from oscillations and overshoot near the constraint boundary due to mismatched update timescales between the policy and the dual variable. CPPO-PID (Stooke et al., 2020) reduces oscillations via PID-controlled multiplier updates, and recent work (Chen et al., 2024) further couple this feedback with multiplier-dependent adaptive primal learning rates to scale the policy update, at the cost of introducing more hyperparameters to tune. We also consider FOCOPS (Zhang et al., 2020) and CUP (Yang et al., 2022), which rely on solving primal–dual subproblems through projection-based updates. Although theoretically motivated, such projections primarily enforce feasibility and do not preserve equivalence with the original constrained optimum, which can degrade empirical performance (Zhang et al., 2023).

**Penalty and augmented penalty methods.** Penalty-based approaches replace the constrained objective with an unconstrained surrogate. IPO (Liu et al., 2020) employs logarithmic barrier functions, requiring feasible initialization and yielding suboptimal solutions in practice. P3O (Zhang et al., 2022) uses an exact $\ell_1$ penalty formulation but relies on sufficiently large penalty coefficients, which can amplify gradient estimation errors. EPO (Gao et al., 2024) employs an adaptive *Exterior point* penalty generated by a seperate Penalty Metric Network. APPO (Dai et al., 2023) introduces a quadratic penalty term to complement the Lagrangian multiplier, improving stability by dampening cost

oscillations. However, these methods typically rely on fixed or globally tuned penalty factors, without explicitly accounting for the local sensitivity of the constraint surface. For an extended discussion on the related works, see Appendix J

## 7. Conclusion and future works

We proposed CSPO, a first-order, primal-dual algorithm for safe policy optimization that incorporates local constraint sensitivity to stabilize feasibility recovery. By scaling the constraint correction using the norm of the constraint gradient, CSPO mitigates overshooting and oscillations near the safety boundary while preserving reward performance, without relying on projections, large penalty coefficients, or second-order optimization. Experiments across navigation and locomotion tasks show that CSPO consistently improves recovery stability and constrained returns compared to strong safe RL baselines. Future work includes extending CSPO to multiple constraints and delayed or noisy cost signals, and generalizing the sensitivity scaling beyond the Euclidean norm using alternative local metrics (e.g Fisher), to incorporate curvature information of the constraint, yielding a geometry-aware extension of CSPO.

## Impact Statement

This paper presents work whose goal is to advance the field of Safe Reinforcement Learning. While improved safety guarantees can benefit real-world decision-making systems, the methods developed here are general-purpose and we do not anticipate any immediate negative societal impacts that

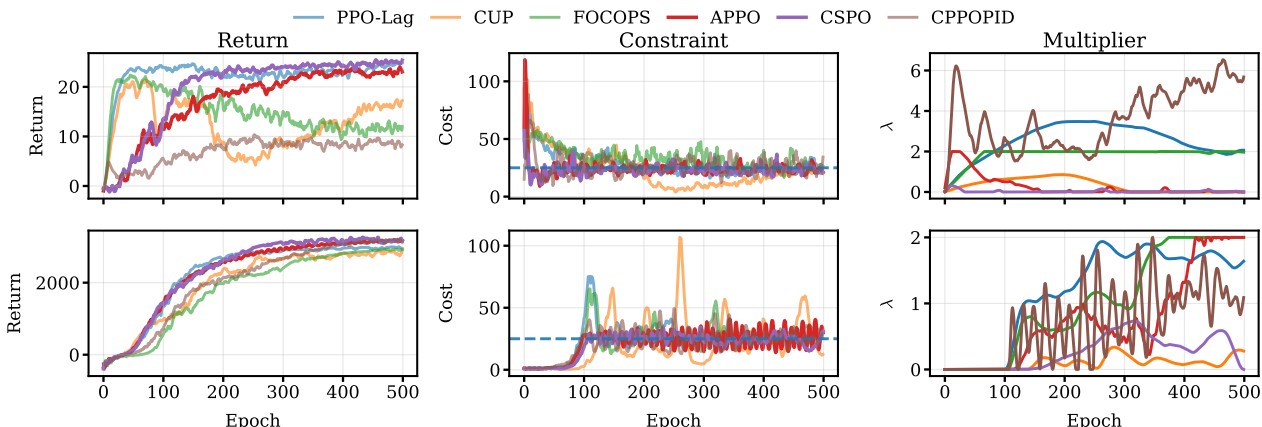

*Figure 3.* Episodic returns and costs for Lagrangian-based algorithms on *PointGoal* (top) and *Ant* (bottom) tasks. CSPO's constraint sensitivity helps mitigating and damping oscillations arising from delayed dual updates.

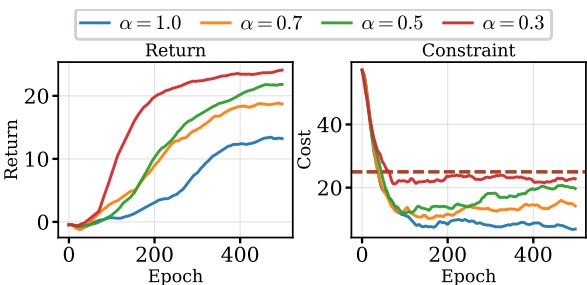

*Figure 4.* Episodic returns and costs of CSPO with different $\alpha$ values on *PointGoal*. Larger $\alpha$ induces more aggressive sensitivity-aware corrections, accelerating feasibility recovery at the expense of reward; smaller values yield conservative safety recovery.

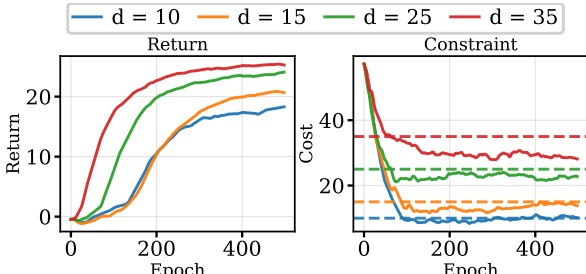

*Figure 5.* Cost-threshold ablation on POINTGOAL. CSPO adapts to increasingly strict cost limits while maintaining stable reward growth and rapid return to feasibility.

require specific discussion.

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

# A. Derivations for Section 3.1

## A.1. Minimum update to the linearized safety boundary

We derive the closed form of the minimal parameter update (in the Euclidean norm) that reaches the *linearized* constraint boundary. Consider the optimization problem:

$$\min_{\Delta\theta} \tfrac{1}{2}\|\Delta\theta\|^2 \quad \text{s.t.} \quad \nabla_\theta g(\theta_k)^\top \Delta\theta + g(\theta_k) = 0 \tag{25}$$

The objective is strictly convex and the constraint is affine, therefore strong duality holds and the KKT conditions are necessary and sufficient for optimality.

From the Lagrangian

$$\mathcal{J}(\Delta\theta, \nu) = \tfrac{1}{2}\|\Delta\theta\|^2 + \nu\big(\nabla_\theta g(\theta_k)^\top \Delta\theta + g(\theta_k)\big),$$

stationarity w.r.t. $\Delta\theta$ gives

$$\nabla_{\Delta\theta}\mathcal{J} = \Delta\theta + \nu\nabla_\theta g(\theta_k) = 0 \quad \Rightarrow \quad \Delta\theta^\star = -\nu\,\nabla_\theta g(\theta_k).$$

Imposing the constraint yields

$$\nabla_\theta g(\theta_k)^\top \Delta\theta^\star = -\nu\,\|\nabla_\theta g(\theta_k)\|^2 = -g(\theta_k),$$

so

$$\nu = \frac{g(\theta_k)}{\|\nabla_\theta g(\theta_k)\|^2}.$$

Therefore,

$$\Delta\theta^\star = -\frac{g(\theta_k)}{\|\nabla_\theta g(\theta_k)\|^2}\,\nabla_\theta g(\theta_k).$$

Finally, the optimal step length is

$$\|\Delta\theta^\star\|^2 = \left(\frac{g(\theta_k)}{\|\nabla_\theta g(\theta_k)\|^2}\right)^2 \|\nabla_\theta g(\theta_k)\|^2 = \frac{g(\theta_k)^2}{\|\nabla_\theta g(\theta_k)\|^2}, \tag{26}$$

which gives

$$\|\Delta\theta^\star\| = \frac{|g(\theta_k)|}{\|\nabla_\theta g(\theta_k)\|}. \tag{27}$$

In the safety-recovery setting, when $g(\theta_k) > 0$, (27) reduces to

$$\|\Delta\theta^\star\| = \frac{g(\theta_k)}{\|\nabla_\theta g(\theta_k)\|}.$$

## A.2. Minimum update for the fractional violation reduction

We derive the minimal update that achieves a fractional reduction of the linearized constraint violation. Let $\sigma \in [0, 1)$ be the desired contraction factor, i.e.,

$$g(\theta_{k+1}) \leq \sigma\, g(\theta_k). \tag{28}$$

Using the first-order approximation

$$g(\theta_k + \Delta\theta) \approx g(\theta_k) + \nabla_\theta g(\theta_k)^\top \Delta\theta, \tag{29}$$

Then consider the optimization problem

$$\min_{\Delta\theta} \tfrac{1}{2}\|\Delta\theta\|^2 \quad \text{s.t.} \quad \nabla_\theta g(\theta_k)^\top \Delta\theta + g(\theta_k) = \sigma\, g(\theta_k).$$

The Lagrangian is

$$\mathcal{J}(\Delta\theta, \nu) = \tfrac{1}{2}\|\Delta\theta\|^2 + \nu\Big(\nabla_\theta g(\theta_k)^\top \Delta\theta + (1 - \sigma)\, g(\theta_k)\Big).$$

Stationarity gives $\Delta\theta + \nu\nabla_\theta g(\theta_k) = 0$, hence

$$\Delta\theta^\star = -\nu\,\nabla_\theta g(\theta_k).$$

Enforcing the constraint yields

$$-\nu\,\|\nabla_\theta g(\theta_k)\|^2 = -(1-\sigma)\,g(\theta_k) \quad \Rightarrow \quad \nu = \frac{(1-\sigma)\,g(\theta_k)}{\|\nabla_\theta g(\theta_k)\|^2}.$$

Therefore

$$\Delta\theta^\star = -\frac{(1-\sigma)\,g(\theta_k)}{\|\nabla_\theta g(\theta_k)\|^2}\,\nabla_\theta g(\theta_k).$$

Finally, the optimal step length is

$$\|\Delta\theta^\star\|^2 = \left(\frac{(1-\sigma)\,g(\theta_k)}{\|\nabla_\theta g(\theta_k)\|^2}\right)^2 \|\nabla_\theta g(\theta_k)\|^2 = \frac{(1-\sigma)^2\,g(\theta_k)^2}{\|\nabla_\theta g(\theta_k)\|^2}. \tag{30}$$

This yields

$$\|\Delta\theta^\star\| = (1-\sigma)\frac{|g(\theta_k)|}{\|\nabla_\theta g(\theta_k)\|}. \tag{31}$$

For the infeasible case $g(\theta_k) > 0$, and setting $\alpha = 1 - \sigma \in [0,1]$, this simplifies to

$$\|\Delta\theta^\star\| = \alpha\frac{g(\theta_k)}{\|\nabla_\theta g(\theta_k)\|},$$

which matches Eq. (11).

## B. Proof of Proposition 3.2.

Assume that the reward objective $L(\theta)$ and the constraint function $g(\theta)$ are continuously differentiable. Consider the original constrained problem

$$\min_\theta\ -L_R(\theta) \quad \text{s.t.} \quad g(\theta) \le 0, \tag{32}$$

and the CSPO-augmented constrained problem

$$\min_\theta\ -L_R(\theta) + q_k(\theta) \quad \text{s.t.} \quad g(\theta) \le 0, \tag{33}$$

where

$$q_k(\theta) = \frac{\alpha}{2}\,w_k\,[g(\theta)]_+^2.$$

Consider the same constraint qualification for both problems. Then the two problems share the same set of KKT solutions.

**Proof.** Let $(\theta^\star, \lambda^\star)$ be a KKT solution of the original problem (32). By primal feasibility, $g(\theta^\star) \le 0$, which implies $[g(\theta^\star)]_+ = 0$ and hence $q_k(\theta^\star) = 0$. Moreover, since the hinge term $[g(\theta)]_+^2$ vanishes identically for all $\theta$ such that $g(\theta) \le 0$, the sensitivity correction term does not contribute to the objective or its first-order variation at any feasible point.

As a result, the stationarity condition of the CSPO-augmented problem reduces to

$$-\nabla_\theta L_R(\theta^\star) + \lambda^\star \nabla_\theta g(\theta^\star) = 0,$$

which coincides exactly with the KKT stationarity condition of the original problem. The remaining KKT conditions—primal feasibility, dual feasibility, and complementary slackness—are identical for both problems. Therefore, $(\theta^\star, \lambda^\star)$ is also a KKT solution of (33).

Conversely, let $(\theta^\star, \lambda^\star)$ be a KKT solution of the CSPO-augmented problem (33). By primal feasibility, $g(\theta^\star) \le 0$, implying $q_k(\theta^\star) = 0$ and $\nabla q_k(\theta^\star) = 0$. Thus, the stationarity condition of the CSPO-augmented problem reduces to that of the original constrained problem, and the remaining KKT conditions coincide. Hence $(\theta^\star, \lambda^\star)$ is a KKT solution of (32).

# C. Convergence analysis

We establish convergence of CSPO's idealized single-step iterates (16) by verifying the conditions of Lin et al. (2020) (Theorem 4.9) and deriving the CSPO-specific rate constants.

The iterates (16) perform two-timescale Gradient Descent Ascent (GDA) on $\min_\theta \max_{\lambda \in \Lambda} \mathcal{L}_k(\theta, \lambda)$, matching Algorithm 1 of Lin et al. (2020) under the correspondence

$$x \leftrightarrow \theta, \quad y \leftrightarrow \lambda, \quad f(x, y) \leftrightarrow \mathcal{L}_k(\theta, \lambda), \quad \eta_x \leftrightarrow \eta_\theta, \quad \eta_y \leftrightarrow \eta_\lambda,$$

with $w_k$ evaluated at the current iterate $\theta_k$ and treated as a constant during the optimization of $\theta$.

**Condition 1: Nonconvex-concave structure.** $\mathcal{L}_k(\theta, \lambda) = -L_R(\theta) + q_k(\theta) + \lambda g(\theta)$ is nonconvex in $\theta$ since $L_R(\theta)$ is generally nonconvex, and linear (hence concave) in $\lambda$. The dual domain $\Lambda = [0, \lambda_{\max}]$ is convex and bounded with diameter $D = \lambda_{\max}$. This satisfies Assumption 4.7 of Lin et al. (2020).

**Condition 2: $L$-smoothness in $\theta$.** For any fixed $\lambda \in \Lambda$, we bound the gradient difference $\|\nabla_\theta \mathcal{L}_k(\theta, \lambda) - \nabla_\theta \mathcal{L}_k(\theta', \lambda)\|$ by decomposing into three terms using (17):

$$\nabla_\theta \mathcal{L}_k(\theta, \lambda) - \nabla_\theta \mathcal{L}_k(\theta', \lambda) = -\big(\nabla L_R(\theta) - \nabla L_R(\theta')\big)$$
$$+ \big(\lambda + \alpha w_k [g(\theta)]_+\big) \nabla g(\theta)$$
$$- \big(\lambda + \alpha w_k [g(\theta')]_+\big) \nabla g(\theta').$$

Adding and subtracting $\big(\lambda + \alpha w_k [g(\theta')]_+\big) \nabla g(\theta)$ and regrouping yields:

$$\nabla_\theta \mathcal{L}_k(\theta, \lambda) - \nabla_\theta \mathcal{L}_k(\theta', \lambda) = \underbrace{-\big(\nabla L_R(\theta) - \nabla L_R(\theta')\big)}_{\text{Term 1}}$$
$$+ \underbrace{\alpha w_k \big([g(\theta)]_+ - [g(\theta')]_+\big) \nabla g(\theta)}_{\text{Term 2}}$$
$$+ \underbrace{\big(\lambda + \alpha w_k [g(\theta')]_+\big)\big(\nabla g(\theta) - \nabla g(\theta')\big)}_{\text{Term 3}}.$$

Applying the triangle inequality and bounding each term:

- $\|\nabla L_R(\theta) - \nabla L_R(\theta')\| \le L_R \|\theta - \theta'\|$ by $L_R$-smoothness of $L_R$.

- $\alpha w_k |[g(\theta)]_+ - [g(\theta')]_+| \|\nabla g(\theta)\| \le \alpha w_{\max} G_g^2 \|\theta - \theta'\|$ since the hinge is 1-Lipschitz, $\|\nabla g\| \le G_g$, and $w_k \le w_{\max}$.

- $(\lambda + \alpha w_k [g(\theta')]_+) \|\nabla g(\theta) - \nabla g(\theta')\| \le (\lambda_{\max} + \alpha w_{\max} B_g) L_g \|\theta - \theta'\|$ by $L_g$-smoothness of $g$ and $[g]_+ \le B_g$.

Summing yields $L$-smoothness with

$$L = L_R + \alpha w_{\max} G_g^2 + (\lambda_{\max} + \alpha w_{\max} B_g) L_g,$$

satisfying the $\ell$-smoothness requirement of Assumption 4.7.

**Condition 3: $G$-Lipschitz in $\theta$.** From (17), uniformly over $\lambda \in \Lambda$:

$$\|\nabla_\theta \mathcal{L}_k(\theta, \lambda)\| \le \underbrace{G_R}_{\|\nabla L_R\|} + \underbrace{(\lambda_{\max} + \alpha w_{\max} B_g) G_g}_{(\lambda + \alpha w_k [g]_+) \|\nabla g\|} =: G,$$

so $\Phi_k(\theta) = \max_{\lambda \in \Lambda} \mathcal{L}_k(\theta, \lambda)$ is $G$-Lipschitz, satisfying the $L$-Lipschitz requirement of Assumption 4.7.

**Conclusion.** All conditions of Theorem 4.9 of Lin et al. (2020) are satisfied with $\ell \leftrightarrow L$, $L$-Lip $\leftrightarrow G$, $D \leftrightarrow \lambda_{\max}$. Applying Theorem 4.9 with step sizes

$$\eta_\theta = \Theta\left(\frac{\varepsilon^4}{L^3 G^2 \lambda_{\max}^2}\right), \qquad \eta_\lambda = \Theta\left(\frac{1}{L}\right)$$

yields the complexity

$$O\left(\frac{L^3 G^2 \lambda_{\max}^2 \Delta_\Phi}{\varepsilon^6} + \frac{L^3 \lambda_{\max}^2 \Delta_0}{\varepsilon^4}\right).$$

Absorbing the initialization-dependent quantities $\Delta_\Phi$ and $\Delta_0$ into constants and retaining the dominant term in $\varepsilon$ gives the simplified rate

$$O\left(\frac{L^3 G^2 \lambda_{\max}^2}{\varepsilon^6}\right).$$

to an $\varepsilon$-stationary point of (16)

## D. Proof of Proposition 4.1

For a fixed episodic iteration $k$ in Algorithm 1, let $L(\theta, \lambda_k)$ denote CSPO's objective in (21), with $w_k$ and $\lambda_k$ held constant throughout the inner loop updates, and assuming that $L(\cdot, \lambda_k)$ is L-smooth in $\theta$. Let $\{\theta_t\}_{t=0}^{T-1}$ be generated by $T$ gradient steps:

$$\theta_{t+1} = \theta_t - \eta \, \nabla_\theta L(\theta_t, \lambda_k), \qquad t = 0, \ldots, T-1,$$

with step size $\eta \in (0, 1/L]$.

By the descent lemma for $L$-smooth functions (Nesterov et al., 2018), for any $\theta$ and $\Delta$,

$$L(\theta + \Delta, \lambda_k) \leq L(\theta, \lambda_k) + \langle \nabla_\theta L(\theta, \lambda_k), \Delta \rangle + \frac{L}{2} \|\Delta\|^2.$$

Applying this inequality at $\theta = \theta_t$ with $\Delta = -\eta \, \nabla_\theta L(\theta_t, \lambda_k)$ yields

$$L(\theta_{t+1}, \lambda_k) \leq L(\theta_t, \lambda_k) - \eta \, \|\nabla_\theta L(\theta_t, \lambda_k)\|^2 + \frac{L\eta^2}{2} \|\nabla_\theta L(\theta_t, \lambda_k)\|^2$$

$$= L(\theta_t, \lambda_k) - \eta\left(1 - \frac{L\eta}{2}\right) \|\nabla_\theta L(\theta_t, \lambda_k)\|^2.$$

Since $\eta \leq 1/L$, we have $1 - \frac{L\eta}{2} \geq \frac{1}{2}$, and therefore

$$L(\theta_{t+1}, \lambda_k) \leq L(\theta_t, \lambda_k) - \frac{\eta}{2} \|\nabla_\theta L(\theta_t, \lambda_k)\|^2.$$

Summing over $t = 0$ to $T-1$ gives

$$\frac{\eta}{2} \sum_{t=0}^{T-1} \|\nabla_\theta L(\theta_t, \lambda_k)\|^2 \leq L(\theta_0, \lambda_k) - L_k(\theta_T, \lambda_k) \leq L(\theta_0, \lambda_k) - L^\star(\lambda_k),$$

where $L^\star(\lambda_k) := \inf_\theta L(\theta, \lambda_k)$. Dividing both sides by $\eta T/2$ yields

$$\frac{1}{T} \sum_{t=0}^{T-1} \|\nabla_\theta L(\theta_t, \lambda_k)\|^2 \leq \frac{2\big(L(\theta_0, \lambda_k) - L^\star(\lambda_k)\big)}{\eta T}.$$

Finally, since $\min_t a_t \leq \frac{1}{T} \sum_t a_t$ for nonnegative $a_t$, we obtain

$$\min_{0 \leq t \leq T-1} \|\nabla_\theta L(\theta_t, \lambda_k)\|^2 \leq \frac{2\big(L(\theta_0, \lambda_k) - L^\star(\lambda_k)\big)}{\eta T} = \mathcal{O}\left(\frac{1}{T}\right),$$

which proves the stationarity rate in Proposition 4.1.

# E. Proof of Proposition 4.2

let $L(\theta, \lambda_k)$ denote CSPO's objective in (21) and consider one inner-loop gradient update

$$\theta_{t+1} = \theta_t - \eta \nabla_\theta L(\theta_t, \lambda_k), \tag{34}$$

with step size $\eta > 0$.

**First-order expansion of the constraint.** Assuming $g$ is continuously differentiable with locally Lipschitz gradient. Then, for sufficiently small $\eta$,

$$\begin{aligned}
g(\theta_{t+1}) &= g(\theta_t) + \nabla g(\theta_t)^\top (\theta_{t+1} - \theta_t) + \mathcal{O}(\|\theta_{t+1} - \theta_t\|^2) \\
&= g(\theta_t) + \nabla g(\theta_t)^\top (\theta_{t+1} - \theta_t) + \mathcal{O}(\eta^2).
\end{aligned} \tag{35}$$

Additionally, assuming the gradients are locally bounded as $\|\nabla L_R(\theta)\| \le G_R$ and $\|\nabla g(\theta)\| \le G_g$.

**Constraint decrease bound.** Let $\theta_t$ be infeasible, i.e., $g(\theta_t) > 0$. Then $[g(\theta_t)]_+ = g(\theta_t)$ and

$$\nabla_\theta \left( \tfrac{\alpha}{2} w [g(\theta)]_+^2 \right) \Big|_{\theta=\theta_t} = \alpha w\, g(\theta_t)\, \nabla g(\theta_t).$$

Differentiating the CSPO's objective at $\theta_t$ yields

$$\nabla_\theta L(\theta_t, \lambda_k) = -\nabla L_R(\theta_t) + \Big( \lambda + \alpha w g(\theta_t) \Big) \nabla g(\theta_t). \tag{36}$$

Substituting (34) into (35) gives

$$g(\theta_{t+1}) = g(\theta_t) - \eta \left\langle \nabla g(\theta_t),\, \nabla_\theta L(\theta_t, \lambda_k) \right\rangle + \mathcal{O}(\eta^2). \tag{37}$$

Using (36) in (37) yields

$$g(\theta_{t+1}) = g(\theta_t) + \eta \left\langle \nabla g(\theta_t),\, \nabla L_R(\theta_t) \right\rangle - \eta\, \lambda \|\nabla g(\theta_t)\|^2 - \eta\, \alpha w\, g(\theta_t) \|\nabla g(\theta_t)\|^2 + \mathcal{O}(\eta^2). \tag{38}$$

Since $\lambda \ge 0$, the term $-\eta\, \lambda \|\nabla g(\theta_t)\|^2$ is non-positive and can be dropped, giving

$$g(\theta_{t+1}) \le g(\theta_t) + \eta \left\langle \nabla g(\theta_t),\, \nabla L_R(\theta_t) \right\rangle - \eta\, \alpha w\, g(\theta_t) \|\nabla g(\theta_t)\|^2 + \mathcal{O}(\eta^2). \tag{39}$$

By Cauchy–Schwarz and the bounded-gradient assumption,

$$\left\langle \nabla g(\theta_t),\, \nabla L_R(\theta_t) \right\rangle \le \|\nabla g(\theta_t)\|\, \|\nabla L_R(\theta_t)\| \le G_g G_R =: \delta.$$

Substituting this bound into (39) yields

$$g(\theta_{t+1}) \le g(\theta_t) - \eta \Big( \alpha w\, g(\theta_t) \|\nabla g(\theta_t)\|^2 - \delta \Big) + \mathcal{O}(\eta^2),$$

which proves Eq. (23).

Finally, a one-step decrease in constraint violation ($g(\theta_{t+1}) < g(\theta_t)$) is ensured whenever the leading-order term is strictly negative, i.e.,

$$\alpha w\, g(\theta_t) \|\nabla g(\theta_t)\|^2 > \delta,$$

and $\eta$ is sufficiently small so that the $\mathcal{O}(\eta^2)$ term does not dominate the first-order decrease. Rearranging gives the sufficient condition

$$g(\theta_t) > \frac{\delta}{\alpha w \|\nabla g(\theta_t)\|^2},$$

completing the proof.

*Table 3.* Training parameters.

| Parameter | CSPO | APPO | P3O | IPO | PPO-Lag | CUP | FOCOPS | TRPOPID | CPPOPID | CPO | PCPO | C-TRPO |
|---|---|---|---|---|---|---|---|---|---|---|---|---|
| Number of hidden layers | 2 | 2 | 2 | 2 | 2 | 2 | 2 | 2 | 2 | 2 | 2 | 2 |
| Number of hidden units | 64 | 64 | 64 | 64 | 64 | 64 | 64 | 64 | 64 | 64 | 64 | 64 |
| Activation function | `tanh` | `tanh` | `tanh` | `tanh` | `tanh` | `tanh` | `tanh` | `tanh` | `tanh` | `tanh` | `tanh` | `tanh` |
| Discount factor $\gamma$ | 0.99 | 0.99 | 0.99 | 0.99 | 0.99 | 0.99 | 0.99 | 0.99 | 0.99 | 0.99 | 0.99 | 0.99 |
| GAE parameter $\lambda^{\text{GAE}}$ | 0.95 | 0.95 | 0.95 | 0.95 | 0.95 | 0.95 | 0.95 | 0.95 | 0.95 | 0.95 | 0.95 | 0.95 |
| Actor learning rate $\eta_\pi$ | 3e-4 | 3e-4 | 3e-4 | 3e-4 | 3e-4 | 3e-4 | 3e-4 | 3e-4 | 3e-4 | N/A | N/A | N/A |
| Critic learning rate $\eta_{V_R}$ | 3e-4 | 3e-4 | 3e-4 | 3e-4 | 3e-4 | 3e-4 | 3e-4 | 3e-4 | 3e-4 | 1e-3 | 1e-3 | 1e-3 |
| Training steps | $10^7$ | $10^7$ | $10^7$ | $10^7$ | $10^7$ | $10^7$ | $10^7$ | $10^7$ | $10^7$ | $10^7$ | $10^7$ | $10^7$ |
| Steps per epoch | 2e4 | 2e4 | 2e4 | 2e4 | 2e4 | 2e4 | 2e4 | 2e4 | 2e4 | 2e4 | 2e4 | 2e4 |
| Update iterations | 10 | 10 | 10 | 10 | 10 | 10 | 10 | 10 | 10 | 10 | 10 | 10 |
| Batch size | 512 | 512 | 512 | 512 | 512 | 512 | 512 | 128 | 512 | 128 | 128 | 256 |
| Penalty factor | N/A | 0.2 | 20 | 10 | N/A | N/A | N/A | N/A | N/A | N/A | N/A | N/A |
| PPO Clip ratio $\epsilon$ | 0.2 | 0.2 | 0.2 | 0.2 | 0.2 | 0.2 | 0.2 | N/A | 0.2 | N/A | N/A | N/A |
| Trust region $[\delta^-, \delta^+]$ | [0,2e-2] | [0,2e-2] | [0,2e-2] | [0,2e-2] | [0,2e-2] | [0,2e-2] | [0,2e-2] | [0,1e-2] | [0,2e-2] | [0,1e-2] | [0,1e-2] | [0,1e-2] |
| cg iterations | N/A | N/A | N/A | N/A | N/A | N/A | N/A | 15 | N/A | 15 | 15 | 10 |
| Damping coefficient | N/A | N/A | N/A | N/A | N/A | N/A | N/A | 0.1 | N/A | 0.1 | 0.1 | 0.1 |
| Backtrack iterations | N/A | N/A | N/A | N/A | N/A | N/A | N/A | 15 | N/A | 15 | 15 | 10 |
| Lagrange multiplier init. | 0.001 | 0.001 | N/A | N/A | 0.001 | 0.001 | 1.0 | N/A | 0.001 | N/A | N/A | N/A |
| Lagrange multiplier learning rate | 0.035 | 0.035 | N/A | N/A | 0.035 | 0.01 | 0.01 | N/A | 0.035 | N/A | N/A | N/A |

*Table 4.* Choice of fractional reduction factor $\alpha$ in each task.

| | Point Goal | Point Button | Car Goal | Car Button | Ant | Humanoid | HalfCheetah | Hopper | Swimmer |
|---|---|---|---|---|---|---|---|---|---|
| $\alpha$ | 0.3 | 0.3 | 0.3 | 0.3 | 0.85 | 0.85 | 0.85 | 0.85 | 0.85 |

**Justification of Remark 4.3.** Assuming the inner-loop iterates remain within a trust region around $\theta_k$, i.e., for a sufficiently small neighborhood radius $\epsilon$: $\|\theta_t - \theta_k\| \leq \varepsilon$, and that $\nabla g$ is $L_g$-Lipschitz on this region. Then

$$\|\nabla g(\theta_t) - \nabla g(\theta_k)\| \leq L_g \varepsilon,$$

which implies

$$\|\nabla g(\theta_k)\| - L_g \varepsilon \ \leq \ \|\nabla g(\theta_t)\| \ \leq \ \|\nabla g(\theta_k)\| + L_g \varepsilon.$$

With $w_k = 1/\|\nabla g(\theta_k)\|^2$, it follows that

$$\frac{(\|\nabla g(\theta_k)\| - L_g \varepsilon)^2}{\|\nabla g(\theta_k)\|^2} \ \leq \ w_k \|\nabla g(\theta_t)\|^2 \ \leq \ \frac{(\|\nabla g(\theta_k)\| + L_g \varepsilon)^2}{\|\nabla g(\theta_k)\|^2}.$$

Hence, for sufficiently small $\varepsilon$, $w_k \|\nabla g(\theta_t)\|^2$ remains close to 1 throughout the inner loop, so the sufficient decrease condition in Proposition 4.2 is well-approximated by the rule of thumb $g(\theta_t) \gtrsim \delta/\alpha$.

## F. Training curves and parameters

In this section we report the training parameters used in the experiments for CSPO and the other baselines, as well as the full training curves. Table 3 shows the hyperparameters used for all baselines, and Table 4 shows the choice of $\alpha$ for CSPO in each task. Training curves are shown in Figure 6 for navigations tasks and Figure 7 for locomotion.

## G. Additional implementation details

**Stabilizing the sensitivity weight $w$.** CSPO uses the sensitivity weight $w_k \propto 1/\|\nabla_\theta \hat{g}(\theta_k)\|^2$ computed from automatic differentiation of the surrogate constraint $\hat{g}$. In practice, raw gradient-norm estimates can be noisy under minibatch sampling and PPO-style clipping. To improve numerical stability, we apply three standard safeguards when computing $w$.

**(i) $\epsilon$-regularization.** We add a small constant $\epsilon > 0$ to avoid division by zero:

$$w_{\text{raw}} = \frac{1}{\|\nabla_\theta \hat{g}(\theta_k)\|^2 + \epsilon},$$

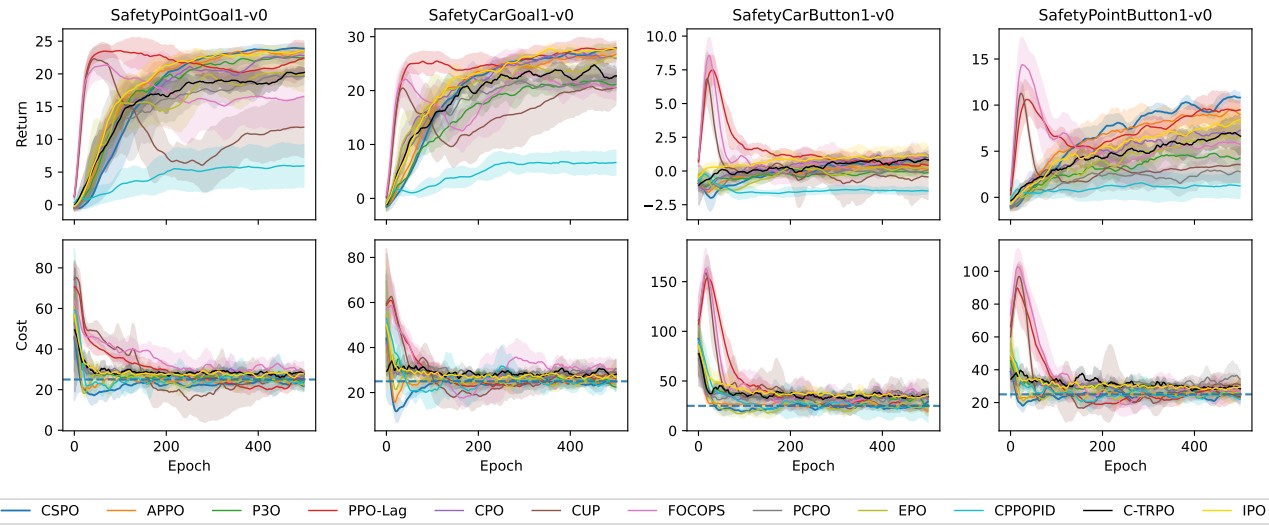

*Figure 6.* Training curves of CSPO and baselines over 5 seeds on 4 Safety Gymnasium navigation tasks. The x-axis shows training epochs, the y-axis return or cost; solid lines denote the mean, shaded areas the standard deviation, and the dashed line the cost threshold (25).

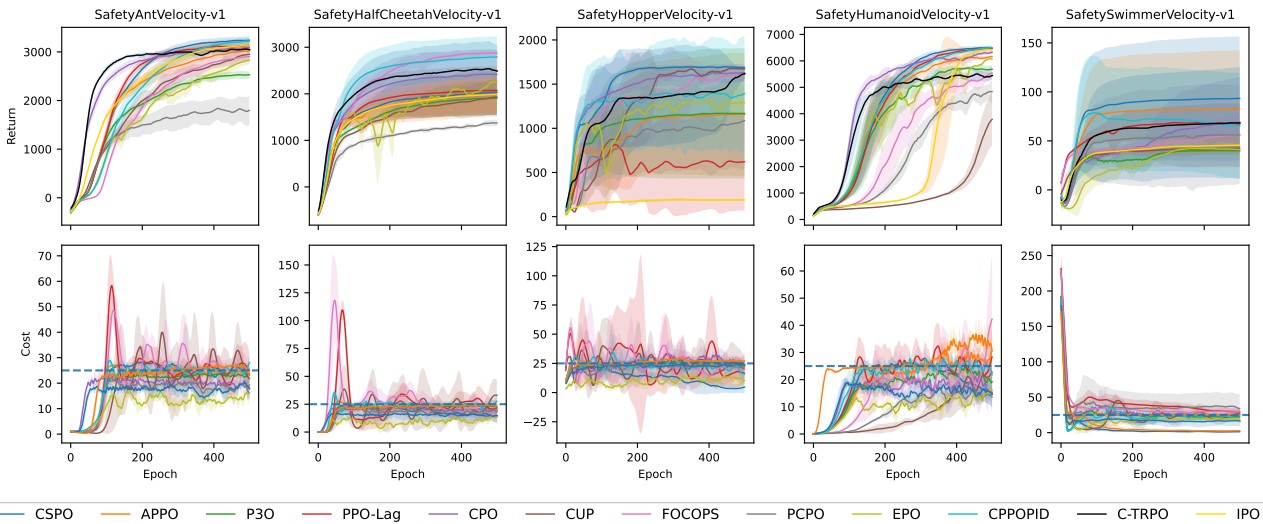

*Figure 7.* Training curves of CSPO and baselines over 5 seeds on 5 Safety Gymnasium locomotion tasks. The x-axis shows training epochs; y-axis return or cost; solid lines denote the mean, shaded areas the standard deviation, and the dashed line the cost threshold (25).

---

**Algorithm 2** Stabilized sensitivity weight computation for CSPO

---

1: **Input:** rollout batch $\tau \sim \pi_{\theta_k}$, cost advantages $\hat{A}^C$, old log-probabilities $\log \pi_{\theta_k}(a|s)$, EMA state $\bar{w}_{k-1}$, smoothing coefficient $\beta \in [0, 1)$, $\epsilon > 0$, clipping bounds $w_{\min}, w_{\max}$.

2: Compute PPO ratio: $r_\theta(s, a) = \exp\left(\log \pi_\theta(a|s) - \log \pi_{\theta_k}(a|s)\right)$

3: Define surrogate constraint-gradient signal : $\hat{g}(\theta) \triangleq \mathbb{E}_{(s,a)\sim\tau}\left[r_\theta(s, a)\, \hat{A}^C(s, a)\right]$

4: Compute constraint gradient via automatic differentiation:
$$g_\theta \leftarrow \nabla_\theta \hat{g}(\theta)\big|_{\theta=\theta_k} = \mathbb{E}_{(s,a)\sim\tau}\left[\hat{A}^C(s, a)\, \nabla_\theta \log \pi_{\theta_k}(a|s)\right]$$

5: Compute raw weight: $w_{\text{raw}} \leftarrow \frac{1}{\|g_\theta\|^2 + \epsilon_w}$

6: Clip weight: $w_{\text{clip}} \leftarrow \text{clip}(w_{\text{raw}}, w_{\min}, w_{\max})$

7: EMA smoothing: $\bar{w}_k \leftarrow \beta \bar{w}_{k-1} + (1 - \beta)\, w_{\text{clip}}$

8: Detach and return stabilized weight: $w_k \leftarrow \text{stopgrad}(\bar{w}_k)$

9: **Output** $w_k$

---

**(ii) Clipping.** We clamp $w_{\mathrm{raw}}$ to a bounded interval to prevent extreme values in very flat or very steep regions:

$$w_{\mathrm{clip}} = \mathrm{clip}(w_{\mathrm{raw}}, w_{\min}, w_{\max}).$$

**(iii) Exponential moving average (EMA).** We further smooth the clipped weight using an EMA to reduce the variance:

$$\bar{w}_k = \beta\,\bar{w}_{k-1} + (1-\beta)\,w_{\mathrm{clip}}, \qquad \beta \in [0, 1).$$

The smoothed weight $\bar{w}_k$ is then used in the policy update, and we *detach* it from the computation graph (i.e., no gradients are backpropagated through $w$), so that the policy update treats $w$ as a constant within the inner loop:

$$w_k := \mathrm{stopgrad}(\bar{w}_k).$$

This ensures that the sensitivity scaling acts as a stable, violation-dependent normalization factor rather than introducing additional higher-order gradient terms. Algorithm 2 summarizes this process.

**Gradient of the surrogate constraint.** The true constraint residual is $J_C(\pi_{\theta_k}) - d$, which is estimated from rollouts but is not differentiable with respect to $\theta$ since it depends on environment transitions. Therefore, we compute $\nabla_\theta \hat{g}(\theta_k)$ from a differentiable PPO-style surrogate that depends on $\theta$ only through the policy ratio $r_\theta(s, a)$. In particular, at $\theta = \theta_k$ we have $r_{\theta_k}(s, a) = 1$ while $\nabla_\theta r_\theta|_{\theta_k} = \nabla_\theta \log \pi_{\theta_k}(a|s)$, yielding a standard policy-gradient form for the surrogate constraint gradient.

## H. Fixed vs adaptive fractional reduction factor $\alpha$

In addition to the fixed-$\alpha$ ablation in Section 5.4, we evaluate a simple violation-adaptive heuristic for the sensitivity coefficient. Motivated by Proposition 4.2, which indicates that the feasibility activation threshold scales as $O(1/\alpha)$, we increase $\alpha$ proportionally to the current constraint violation. Specifically, at epoch $k$ we define $\phi_k = J_C(\pi_{\theta_k}) - d$ and set $\alpha_k = \mathrm{clip}([\phi_k]_+/(d + \varepsilon), 0, 1)$. Figure 8 compares fixed $\alpha = 0.3$ against the adaptive schedule on 4 navigation tasks and Figure 9 compares fixed $\alpha = 0.85$ against the adaptive schedule on 4 locomotion tasks. Overall, adaptive $\alpha$ yields comparable feasibility behavior, but does not consistently improve reward performance over a well-tuned fixed $\alpha$.

## I. Sensitivity to dual hyperparameters

We analyze the sensitivity of CSPO to the initialization and learning rate of the Lagrange multiplier and compare it against PPO-Lag on representative navigation and locomotion tasks. Learning curves are shown in Figures 10 - 11.

**Initialization of the multiplier.** Varying the initial multiplier value $\lambda_0$ (Figure 10), we observe that for CSPO, large initial values (e.g., $\lambda_0 = 0.1$) induce conservative early behavior, resulting in lower returns and persistent under-utilization of the safety budget. In contrast, smaller values ($\lambda_0 \in [0.001, 0.05]$) consistently converge to similar final returns and constraint satisfaction levels. PPO-Lag exhibits markedly higher sensitivity: large $\lambda_0$ often leads to overly conservative policies, while small $\lambda_0$ can cause prolonged constraint violations or oscillatory cost behavior. These results indicate that CSPO recovers more reliably from unfavorable multiplier initialization, reflecting improved robustness rather than complete insensitivity.

**Learning rate of the multiplier.** We further vary the multiplier learning rate $\eta_\lambda$ (Figure 11). CSPO remains stable across a broad range of values, with changes in $\eta_\lambda$ primarily affecting transient dynamics rather than final performance. In contrast, PPO-Lag shows stronger coupling between $\eta_\lambda$ and training stability, manifesting as larger cost oscillations and increased sensitivity to tuning. This behavior is consistent with the design of CSPO: sensitivity-aware correction absorbs part of the feasibility restoration that would otherwise rely solely on the dual update, reducing the burden on the multiplier dynamics and improving robustness to its step size.

## J. Extended Related works

### J.1. Preliminary

We briefly summarize the CMDP formulation and the constrained RL problem discussed in Section 2. A CMDP augments an MDP with $m$ cost functions $\{c_i\}_{i=1}^m$ and corresponding limits $\{d_i\}_{i=1}^m$. For a stationary policy $\pi$, the discounted cost

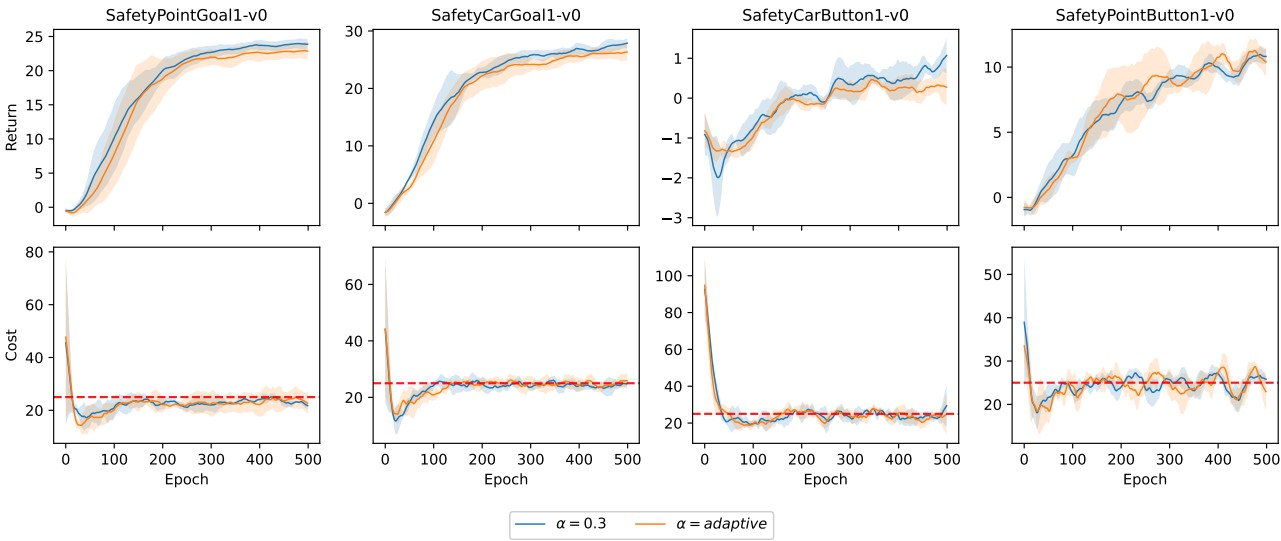

*Figure 8.* Fixed vs. adaptive $\alpha$ in CSPO on four navigation tasks across 5 seeds. We compare CSPO with a fixed ($\alpha = 0.3$) against a simple violation-adaptive schedule (Appendix H). Top: episodic return. Bottom: episodic cost with the cost limit shown as a dashed line. Curves show mean across seeds and shaded regions indicate variability.

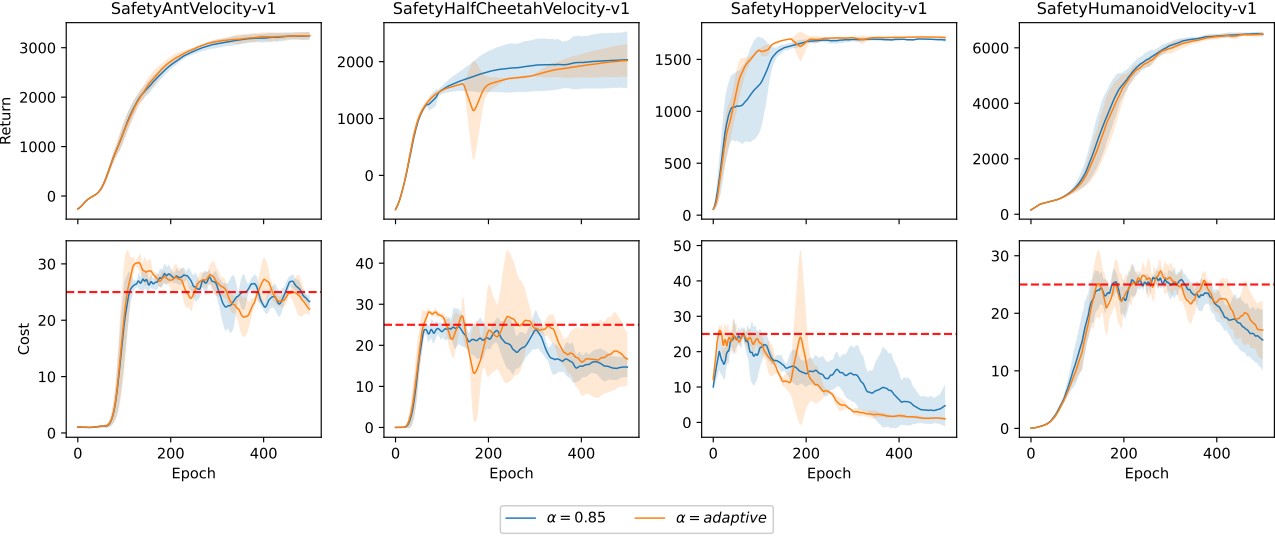

*Figure 9.* Fixed vs. adaptive $\alpha$ in CSPO on four locomotion tasks across 5 seeds. We compare CSPO with a fixed ($\alpha = 0.3$) against a simple violation-adaptive schedule (Appendix H). Top: episodic return. Bottom: episodic cost with the cost limit shown as a dashed line. Curves show mean across seeds and shaded regions indicate variability.

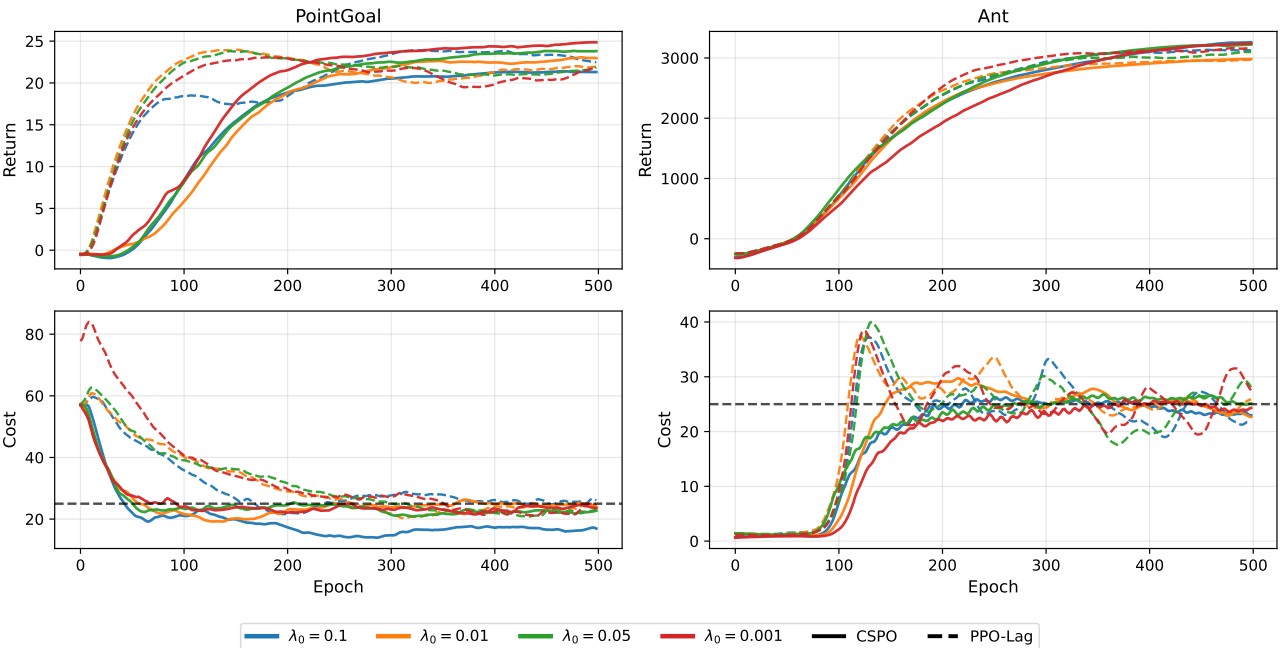

*Figure 10.* Training curves of CSPO vs PPO-Lag across four different initializations of the Lagrange multiplier $\lambda$.

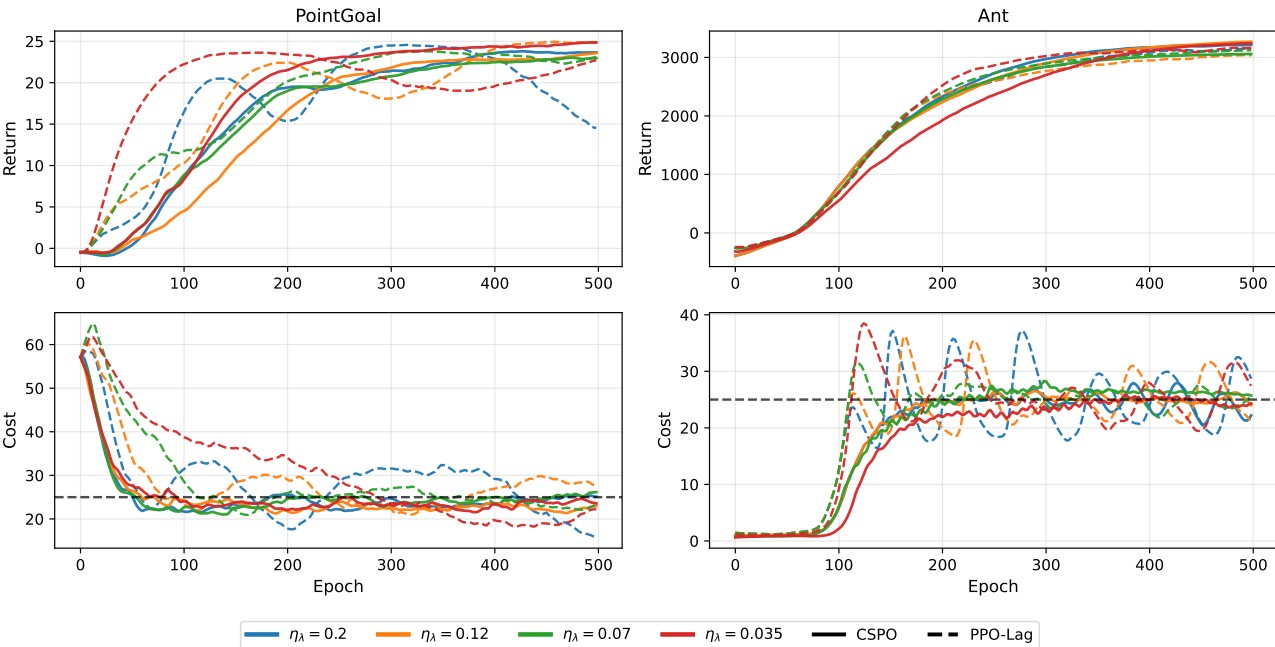

*Figure 11.* Training curves of CSPO vs PPO-Lag across four different learning rates $\eta$ for the Lagrange multiplier $\lambda$.

*Table 5.* Recovery metrics across algorithms and environments. TTS: time to safety (epochs). RP: reward preservation ($J_{\text{after}}/J_{\text{before}}$). #V: Number of violations. Values are mean $\pm$ std across seeds.

| Environment | | CSPO | PPO-Lag | APPO | FOCOPS | CUP | CPPOPID |
|---|---|---|---|---|---|---|---|
| PointGoal | TTS | **3.098 ± 0.421** | 7.110 ± 3.081 | 4.089 ± 0.383 | 21.452 ± 18.541 | 9.336 ± 8.728 | 6.477 ± 1.515 |
| | RP | 1.000 ± 0.003 | 1.002 ± 0.014 | 0.996 ± 0.003 | 0.9889 ± 0.0074 | 0.9792 ± 0.0361 | 0.9928 ± 0.0166 |
| | #V | 66.4 ± 23.29 | 111.6 ± 30.63 | 116.33 ± 26.83 | 214.5 ± 36.37 | 98.2 ± 109.33 | 143.6 ± 21.89 |
| PointButton | TTS | **3.703 ± 0.411** | 5.144 ± 2.14 | 5.155 ± 0.695 | 9.479 ± 5.699 | 12.208 ± 5.236 | 5.041 ± 2.158 |
| | RP | 0.961 ± 0.014 | 0.980 ± 0.016 | 0.942 ± 0.007 | 0.9984 ± 0.0073 | 0.9732 ± 0.0238 | 1.0132 ± 0.0934 |
| | #V | 104.0 ± 20.02 | 82.6 ± 28.27 | 145.0 ± 11.37 | 130.2 ± 17.61 | 126.0 ± 16.05 | 124.6 ± 14.81 |
| CarGoal | TTS | **4.204 ± 0.611** | 7.240 ± 1.850 | 4.382 ± 0.651 | 39.515 ± 42.679 | 9.872 ± 7.085 | 6.131 ± 0.862 |
| | RP | 1.000 ± 0.006 | 1.004 ± 0.006 | 0.994 ± 0.004 | 1.0318 ± 0.0684 | 1.0098 ± 0.0229 | 0.9932 ± 0.0171 |
| | #V | 116.8 ± 13.718 | 141.2 ± 16.05 | 134.6 ± 19.32 | 200.8 ± 17.61 | 117.6 ± 15.65 | 148.0 ± 26.07 |
| CarButton | TTS | **4.525 ± 0.270** | 11.834 ± 3.46 | 4.977 ± 0.626 | 11.372 ± 1.951 | 10.367 ± 5.576 | 12.260 ± 2.338 |
| | RP | 1.005 ± 0.347 | 0.850 ± 0.259 | 1.051 ± 0.3247 | 0.800 ± 0.08 | 0.8464 ± 0.4363 | 0.8264 ± 0.0519 |
| | #V | 116.33 ± 11.67 | 214.2 ± 27.97 | 141.8 ± 19.12 | 209.5 ± 36.24 | 160.5 ± 10.14 | 107.6 ± 13.50 |
| Ant | TTS | 5.151 ± 0.954 | 14.045 ± 3.210 | **4.591 ± 0.304** | 23.988 ± 4.726 | 21.388 ± 5.019 | 8.870 ± 0.398 |
| | RP | 1.001 ± 0.001 | 0.9979 ± 0.0038 | 0.9975 ± 0.0010 | 1.0330 ± 0.0130 | 1.0229 ± 0.0202 | 1.0046 ± 0.0041 |
| | #V | 122.6 ± 22.04 | 122.0 ± 17.79 | 138.6 ± 8.20 | 150.4 ± 41.53 | 124.8 ± 13.10 | 134.8 ± 14.72 |
| Humanoid | TTS | 4.867 ± 0.524 | 9.690 ± 1.706 | **4.188 ± 0.750** | 8.139 ± 2.425 | 6.267 ± 1.525 | 6.479 ± 0.318 |
| | RP | 1.006 ± 0.005 | 1.0065 ± 0.0074 | 1.0001 ± 0.0009 | 1.0282 ± 0.0197 | 1.0575 ± 0.0590 | 1.0046 ± 0.0051 |
| | #V | 101.00 ± 23.56 | 140.4 ± 25.30 | 135.4 ± 11.39 | 79.00 ± 11.23 | 36.00 ± 5.71 | 124.0 ± 10.93 |
| HalfCheetah | TTS | 8.229 ± 3.949 | 26.290 ± 15.492 | **3.206 ± 0.821** | 35.733 ± 19.674 | 30.533 ± 3.620 | 9.323 ± 2.197 |
| | RP | 1.003 ± 0.002 | 1.0075 ± 0.0045 | 0.9986 ± 0.0025 | 1.0249 ± 0.0286 | 1.0202 ± 0.0199 | 1.0018 ± 0.0015 |
| | #V | 40.25 ± 22.11 | 110.2 ± 19.17 | 83.8 ± 33.40 | 116.88 ± 12.53 | 90.20 ± 17.13 | 118.4 ± 4.77 |
| Hopper | TTS | **2.333 ± 1.040** | 20.656 ± 14.980 | 2.961 ± 1.373 | 14.111 ± 10.238 | 23.683 ± 8.318 | 9.479 ± 1.506 |
| | RP | 1.002 ± 0.002 | 0.8188 ± 0.1797 | 0.9966 ± 0.0104 | 1.0041 ± 0.0093 | 1.0678 ± 0.1872 | 0.9337 ± 0.1296 |
| | #V | 13.333 ± 5.77 | 83.2 ± 43.25 | 129.2 ± 39.25 | 110.8 ± 17.28 | 90.8 ± 8.31 | 114.22 ± 13.34 |
| Swimmer | TTS | **4.387 ± 2.030** | 8.656 ± 2.002 | 5.214 ± 2.120 | 77.500 ± 76.062 | 38.450 ± 20.613 | 12.796 ± 2.642 |
| | RP | 0.9106 ± 0.9853 | 0.811 ± 0.179 | 1.001 ± 0.002 | 1.0135 ± 0.1580 | 1.0065 ± 0.0082 | 0.9915 ± 0.0235 |
| | #V | 20.10 ± 5.35 | 28.0 ± 2.175 | 25.45 ± 4.45 | 50.33 ± 16.35 | 93.60 ± 11.38 | 116.45 ± 17.37 |

return is

$$J_{c_i}(\pi) = \mathbb{E}_{\tau \sim \pi} \left[ \sum_{t=0}^{\infty} \gamma^t c_i(s_t, a_t, s_{t+1}) \right], \tag{40}$$

and the feasible policy set is

$$\Pi_{\text{safe}} = \{\pi \in \Pi : J_{c_i}(\pi) \leq d_i, \ \forall i\}. \tag{41}$$

Safe reinforcement learning aims to maximize return subject to feasibility, $\pi^\star = \arg\max_{\pi \in \Pi_{\text{safe}}} J(\pi)$.

Let $d^\pi$ denote the normalized discounted state-visitation distribution of $\pi$. For any bounded function $f$, the performance difference lemma (Kakade & Langford, 2002) gives

$$J_f(\pi') - J_f(\pi) = \frac{1}{1-\gamma} \mathbb{E}_{s \sim d^{\pi'}, \, a \sim \pi'} \left[ A_f^\pi(s, a) \right], \tag{42}$$

Applying (42) to the reward $r$ and each cost $c_i$ allows rewriting the safe RL objective as an iterative policy search. For parametrized stochastic policies $\pi_\theta(a \mid s)$ with $\theta \in \mathbb{R}^d$ and a current policy $\pi_{\theta_k}$, the updated policy $\pi_{\theta_{k+1}}$ is obtained by maximizing the reward advantage while satisfying the cost constraints

$$\pi_{\theta_{k+1}} = \arg\max_{\pi_\theta} \mathbb{E}_{s \sim d_{\pi_\theta}, \, a \sim \pi_\theta} \left[ A_r^{\pi_{\theta_k}}(s, a) \right] \tag{43}$$

$$\text{s.t.} \quad J_{c_i}(\pi_{\theta_k}) + \frac{1}{1-\gamma} \mathbb{E}_{s \sim d_{\pi_\theta}, \, a \sim \pi_\theta} \left[ A_{c_i}^{\pi_{\theta_k}}(s, a) \right] \leq d_i, \qquad \forall i.$$

This formulation provides the common basis for primal–dual and trust-region methods for CMDPs, and will be used to analyze constraint violation dynamics and safety recovery behavior in later sections.

## J.2. Classical Primal-Dual Methods for CMDPs

We recall the standard primal–dual approach for constrained reinforcement learning. We consider parameterized stochastic policies $\pi_\theta$ and a single cost constraint $J_c(\theta) \leq d$. Defining the constraint function $g(\theta) := J_c(\theta) - d$, and the reward objective $J(\theta) = \mathbb{E}_{s \sim d_{\pi_\theta}, a \sim \pi_\theta}\left[A_r^{\pi_{\theta_k}}(s, a)\right]$ the constrained optimization problem can be written as

$$\min_\theta \ -J(\theta) \quad \text{s.t.} \quad g(\theta) \leq 0.$$

The associated Lagrangian is

$$\mathcal{L}(\theta, \lambda) = -J(\theta) + \lambda\, g(\theta), \qquad \lambda \geq 0. \tag{44}$$

Classical primal–dual methods seek a saddle point of (44) by alternating updates of the primal variable $\theta$ and the dual variable $\lambda$. Using first-order methods, the updates take the form

$$\theta_{k+1} = \theta_k - \eta_\theta\Big(\nabla_\theta J(\theta_k) - \lambda_k \nabla_\theta g(\theta_k)\Big), \tag{45}$$

$$\lambda_{k+1} = \big[\lambda_k + \eta_\lambda g(\theta_k)\big]_+, \tag{46}$$

where $\eta_\theta, \eta_\lambda > 0$ are step sizes and $[\cdot]_+$ denotes projection onto the nonnegative orthant.

In this formulation, the dual variable $\lambda$ accumulates constraint violation over iterations: when $g(\theta_k) > 0$, $\lambda$ increases and strengthens the influence of the constraint gradient in subsequent parameter updates. Constraint satisfaction is therefore enforced indirectly through the growth of $\lambda$, which balances reward minimization and feasibility.

This parameter-space primal–dual scheme constitutes the canonical first-order approach for CMDPs and underlies many safe reinforcement learning algorithms, including PPO-Lag (Ray et al., 2019) and related methods.

## J.3. Augmented Lagrangian Methods for CMDPs

Augmented Lagrangian methods extends primal-dual by adding a quadratic deviation penalty to the objective. A standard approach is to convert the inequality constraint into equality by introducing a nonnegative slack variable $h(\theta) = g(\theta) + s, s \geq 0$ Consider a CMDP with a single expected cost constraint The augmented Lagrangian associated with this equality-constrained problem is

$$\begin{aligned}
\mathcal{L}_\rho(\theta, s, \lambda) &= -J(\theta) + \lambda h(\theta) + \frac{\rho}{2} h(\theta)^2 \\
&= -J(\theta) + \lambda\big(g(\theta) + s\big) + \frac{\rho}{2}\big(g(\theta) + s\big)^2
\end{aligned} \tag{47}$$

where $\lambda \in \mathbb{R}$ is the Lagrange multiplier and $\rho > 0$ is a penalty parameter.

Minimizing (47) with respect to the slack variable admits a closed-form solution,

$$s^\star(\theta, \lambda) = \big[-g(\theta) - \tfrac{\lambda}{\rho}\big]_+.$$

Substituting $s^\star$ back into (47) yields the Powell–Hestenes–Rockafellar (PHR) form of the augmented Lagrangian,

$$\mathcal{L}_\rho(\theta, \lambda) = -J(\theta) + \frac{1}{2\rho}\left(\big[\lambda + \rho g(\theta)\big]_+^2 - \lambda^2\right), \tag{48}$$

which penalizes constraint violations through a squared hinge term.

The corresponding gradient with respect to the policy parameters $\theta$ takes the form

$$\nabla_\theta \mathcal{L}_\rho(\theta, \lambda) = \nabla_\theta J(\theta) - \mathbf{1}\Big\{g(\theta) \geq -\tfrac{\lambda}{\rho}\Big\}\big(\lambda + \rho g(\theta)\big)\nabla_\theta g(\theta), \tag{49}$$

showing that augmented Lagrangian methods effectively activate the quadratic penalty early before violation happens, when the constraint $g(\theta)$ exceeds the $-\frac{\lambda}{\rho}$ threshold. Yielding a dynamic safety threshold that changes whenever the dual variable $\lambda$ and the penalty factor $\rho$ are updated. The dual variable is typically updated via,

$$\lambda_{k+1} = \left[\lambda_k + \rho g(\theta_k)\right]_+, \tag{50}$$

so that the same penalty parameter $\rho$ governs both the magnitude of the primal correction and the growth of the multiplier. Additionally, the penalty factor is also iteratively increased whenever the constraint violation does not decrease

Augmented Lagrangian methods improve upon pure penalty approaches by preserving the original constrained optimum and, under suitable conditions, admitting exact solutions with finite $\rho$. However, in stochastic and nonconvex settings such as deep reinforcement learning, coupling the primal correction and the dual update through a shared penalty parameter can lead to aggressive multiplier growth and oscillatory behavior, often requiring careful task-dependent tuning of $\rho$.

### J.4. Penalty Methods for CMDPs

Penalty methods enforce the constraint by incorporating it directly into the objective through a penalty term, yielding the unconstrained problem

$$\begin{aligned}
\mathcal{P}_\rho(\theta) &= -J(\theta) + \rho\,\phi\big(g(\theta)\big) \\
&= -J(\theta) + \rho\,\phi\big(J_c(\theta) - d\big),
\end{aligned} \tag{51}$$

where $\rho > 0$ is a penalty coefficient and $\phi : \mathbb{R} \to \mathbb{R}_+$ is a nonnegative penalty function satisfying $\phi(g) = 0$ for $g \leq 0$. Theoretical guarantees of penalty methods depend on the choice of $\phi$. For smooth penalties such as quadratic functions, exact equivalence with the original constrained problem typically requires $\rho \to \infty$. Finite exactness can be achieved only for specific nonsmooth penalties under additional regularity conditions. As a result, penalty methods often require large or problem-dependent penalty coefficients to enforce feasibility, which can amplify gradient variance and lead to overly conservative behavior in stochastic and nonconvex settings such as deep reinforcement learning.

### J.5. Related Safe RL Algorithms

In this section we detail the optimization process and objective of the related safe RL algorithms discussed in Section 6, being categorized into primal-dual, augmented lagrangian, penalty and second-order trust region methods.

#### J.5.1. CPO (ACHIAM ET AL., 2017)

Constrained Policy Optimization (CPO) formulates policy updates as a trust-region constrained optimization problem. Given the current policy $\pi_{\theta_k}$, CPO computes $\pi_{\theta_{k+1}}$ by solving

$$\begin{aligned}
\max_{\pi_\theta \in \Pi_\theta} \;\; & \mathbb{E}_{s \sim d_{\pi_{\theta_k}}, a \sim \pi_\theta}\left[A_r^{\pi_{\theta_k}}(s,a)\right] \\
\text{s.t. } & J_c(\pi_{\theta_k}) + \frac{1}{1-\gamma}\mathbb{E}_{s \sim d_{\pi_{\theta_k}}, a \sim \pi_\theta}\left[A_c^{\pi_{\theta_k}}(s,a)\right] \leq d, \\
& \bar{D}_{\mathrm{KL}}(\pi_\theta \,\|\, \pi_{\theta_k}) \leq \delta.
\end{aligned} \tag{52}$$

Since (52) is intractable, CPO applies local approximations. Using first-order Taylor expansions of the reward and cost objectives and a second-order approximation of the KL constraint, the update is approximated as the quadratic program

$$\begin{aligned}
\max_{\theta} \;\; & (\theta - \theta_k)^\top g \\
\text{s.t. } & (\theta - \theta_k)^\top a \leq -c, \\
& (\theta - \theta_k)^\top H (\theta - \theta_k) \leq \delta,
\end{aligned} \tag{53}$$

where $g = \nabla_\theta J(\theta_k)$, $a = \nabla_\theta J_c(\theta_k)$, $c = J_c(\pi_{\theta_k}) - d$, and $H$ is the Fisher information matrix associated with the KL divergence.

Problem (53) admits a closed-form solution via its dual. Let $(\lambda^\star, \nu^\star)$ denote the optimal dual variables associated with the trust-region and cost constraints, respectively. If the linearized problem is feasible, the update direction is

$$\theta_{k+1} = \theta_k + \frac{1}{\lambda^\star} H^{-1}(g - \nu^\star a). \tag{54}$$

Otherwise, the update follows the boundary of the cost constraint,

$$\theta_{k+1} = \theta_k - \sqrt{\frac{2\delta}{a^\top H^{-1}a}}\, H^{-1}a. \tag{55}$$

The required matrix-vector products with $H^{-1}$ are computed using conjugate gradient methods, yielding a natural-gradient step that enforces feasibility at each iteration.

### J.5.2. PCPO (YANG ET AL., 2020)

Projection-Based Constrained Policy Optimization (PCPO) reformulates constrained policy optimization as a two-stage procedure consisting of a reward improvement step followed by a projection onto the feasible set.

Given the current policy $\pi_{\theta_k}$, PCPO first computes an intermediate policy by solving the unconstrained trust-region problem

$$\pi_{\theta_{k+\frac{1}{2}}} = \arg \max_{\pi_\theta \in \Pi_\theta} \mathbb{E}_{s \sim d_{\pi_{\theta_k}},\, a \sim \pi_\theta} \left[ A_r^{\pi_{\theta_k}}(s, a) \right]$$
$$\text{s.t. } \bar{D}_{\mathrm{KL}}(\pi_\theta \,\|\, \pi_{\theta_k}) \le \delta. \tag{56}$$

If the resulting policy violates the cost constraint, PCPO performs a projection step to restore feasibility:

$$\pi_{\theta_{k+1}} = \arg \min_{\pi_\theta \in \Pi_\theta} D(\pi_\theta, \pi_{\theta_{k+\frac{1}{2}}})$$
$$\text{s.t. } J_c(\pi_{\theta_k}) + \frac{1}{1-\gamma} \mathbb{E}_{s \sim d_{\pi_{\theta_k}},\, a \sim \pi_\theta} \left[ A_c^{\pi_{\theta_k}}(s, a) \right] \le d, \tag{57}$$

where $D(\cdot, \cdot)$ is a chosen divergence measure.

Applying the same local approximations as in CPO—first-order expansions of the reward and cost objectives and a quadratic approximation of the trust-region constraint—yields the update

$$\theta_{k+1} = \theta_k - \sqrt{\frac{2\delta}{g^\top H^{-1}g}}\, H^{-1}g - \max\left(0, \frac{g^\top H^{-1}a + c}{a^\top L^{-1}a}\right) L^{-1}a, \tag{58}$$

where $g = \nabla_\theta J(\theta_k)$, $a = \nabla_\theta J_c(\theta_k)$, $c = J_c(\pi_{\theta_k}) - d$, $H$ is the Fisher information matrix, and $L = H$ when $D$ is the KL divergence (or $L = I$ for an $\ell_2$ metric).

Unlike CPO, which solves a joint saddle-point problem, PCPO decouples reward optimization and constraint satisfaction via explicit projection, simplifying the update at the cost of enforcing feasibility only after the reward step.

### J.5.3. C-TRPO (MILOSEVIC ET AL., 2025)

Constrained Trust Region Policy Optimization (C-TRPO) extends trust-region policy optimization by embedding safety directly into the trust region itself. Rather than enforcing constraints through penalties or dual variables, C-TRPO controls constraint satisfaction via a constraint-aware divergence.

Given a safe policy $\pi_k \in \Pi_{\text{safe}}$, C-TRPO computes the next policy by solving

$$\pi_{k+1} = \arg \max_{\pi \in \Pi} \mathcal{A}_r^{\pi_k}(\pi) \quad \text{s.t.} \quad \bar{D}_C(\pi \| \pi_k) \le \delta, \tag{59}$$

where $\mathcal{A}_r^{\pi_k}(\pi)$ denotes the expected reward advantage and $\bar{D}_C$ is a surrogate constraint divergence.

The surrogate divergence is defined as

$$\bar{D}_C(\pi \| \pi_k) = \bar{D}_{\mathrm{KL}}(\pi \| \pi_k) + \beta\, \bar{D}_\phi(\pi \| \pi_k), \tag{60}$$

where $\bar{D}_{\mathrm{KL}}$ is the standard discounted KL divergence and $\bar{D}_\phi$ is a Bregman-type divergence derived from the cost advantage $A_c^{\pi_k}(\pi)$, approximating changes in expected cost to first order. The parameter $\beta > 0$ controls the influence of the constraint on the trust region geometry.

The optimization in (59) is solved using standard TRPO-style approximations: a linearization of the objective and a quadratic approximation of the divergence, leading to a second-order update computed via conjugate gradients.

If the current policy is unsafe, C-TRPO switches to a recovery step that minimizes the cost under a standard KL trust region. Overall, C-TRPO can be interpreted as a trust-region method that incorporates constraint information directly into the policy-space geometry, yielding conservative but stable updates near the constraint boundary.

### J.5.4. FOCOPS (ZHANG ET AL., 2020)

First-Order Constrained Optimization in Policy Space (FOCOPS) formulates safe policy optimization directly in the non-parameterized policy space, avoiding second-order approximations while enforcing constraints via a primal–dual structure.

Given the current policy $\pi_{\theta_k}$, FOCOPS first solves the constrained optimization problem in policy space

$$
\begin{aligned}
\pi^\star = \arg \max_{\pi \in \Pi} \; & \mathbb{E}_{s \sim d_{\pi_{\theta_k}}, a \sim \pi} \left[ A_r^{\pi_{\theta_k}}(s,a) \right] \\
\text{s.t. } & J_c(\pi_{\theta_k}) + \frac{1}{1-\gamma} \mathbb{E}_{s \sim d_{\pi_{\theta_k}}, a \sim \pi} \left[ A_c^{\pi_{\theta_k}}(s,a) \right] \le d, \\
& \bar{D}_{\mathrm{KL}}(\pi \,\|\, \pi_{\theta_k}) \le \delta.
\end{aligned}
\tag{61}
$$

When $\pi_{\theta_k}$ is feasible, the solution in policy space admits a closed-form expression:

$$
\pi^\star(a|s) = \frac{\pi_{\theta_k}(a|s)}{Z_{\lambda,\nu}(s)} \exp \left( \frac{1}{\lambda} \left( A_r^{\pi_{\theta_k}}(s,a) - \nu A_c^{\pi_{\theta_k}}(s,a) \right) \right),
\tag{62}
$$

where $Z_{\lambda,\nu}(s)$ is the normalizing partition function, and $\lambda, \nu \ge 0$ are dual variables obtained by solving

$$
\min_{\lambda,\nu \ge 0} \; \lambda\nu + \nu\tilde{b} + \lambda \mathbb{E}_{s \sim d_{\pi_{\theta_k}}, a \sim \pi^\star} \left[ \log Z_{\lambda,\nu}(s) \right], \qquad \tilde{b} = (1-\gamma)(d - J_c(\pi_{\theta_k})).
\tag{63}
$$

Since $\pi^\star$ generally lies outside the parameterized policy class $\Pi_\theta$, FOCOPS performs a projection step back to the parametric space:

$$
\theta_{k+1} = \arg \min_\theta \mathbb{E}_{s \sim d_{\pi_{\theta_k}}} \left[ \mathrm{KL} \left( \pi_\theta(\cdot|s) \,\|\, \pi^\star(\cdot|s) \right) \right].
\tag{64}
$$

FOCOPS thus implements a first-order primal–dual update entirely in policy space, replacing trust-region constraints with KL-regularized exponentiated updates and avoiding second-order optimization.

### J.5.5. CUP (YANG ET AL., 2022)

Constrained Update Projection (CUP) follows a two-step optimization scheme that alternates between a regularized policy improvement step and a constraint-aware projection step. Unlike second-order trust-region methods, CUP relies only on first-order information while explicitly enforcing feasibility.

**Policy improvement.**   Given the current policy $\pi_{\theta_k}$, CUP first computes an intermediate policy $\pi_{\theta_{k+\frac{1}{2}}}$ by solving

$$
\pi_{\theta_{k+\frac{1}{2}}} = \arg \max_{\pi_\theta \in \Pi_\theta} \left\{ \mathbb{E}_{s \sim d_{\pi_{\theta_k}}^\lambda, a \sim \pi_\theta} \left[ A_{\mathrm{GAE}}^{\pi_{\theta_k}}(s,a) \right] - \alpha_q \mathbb{E}_{s \sim d_{\pi_{\theta_k}}^\lambda} \left[ \mathrm{KL}(\pi_{\theta_k}(\cdot|s) \,\|\, \pi_\theta(\cdot|s)) \right] \right\},
\tag{65}
$$

where $A_{\mathrm{GAE}}^{\pi_{\theta_k}}$ denotes the generalized advantage estimate, $d_{\pi_{\theta_k}}^\lambda$ is the $\lambda$-discounted state visitation distribution, and $\alpha_q > 0$ controls the KL regularization strength.

**Projection.**   The intermediate policy is then projected back onto the constraint set by solving

$$
\begin{aligned}
\pi_{\theta_{k+1}} = \arg \min_{\pi_\theta \in \Pi_\theta} \; & D \left( \pi_\theta, \pi_{\theta_{k+\frac{1}{2}}} \right) \\
\text{s.t. } & J_c(\pi_{\theta_k}) + \frac{1}{1-\tilde{\gamma}} \mathbb{E}_{s \sim d_{\pi_{\theta_k}}^\lambda, a \sim \pi_\theta} \left[ A_{\mathrm{GAE},c}^{\pi_{\theta_k}}(s,a) \right] \\
& + \beta_q \mathbb{E}_{s \sim d_{\pi_{\theta_k}}^\lambda} \left[ \mathrm{KL}(\pi_{\theta_k}(\cdot|s) \,\|\, \pi_\theta(\cdot|s)) \right] \le d,
\end{aligned}
\tag{66}
$$

where $A_{\mathrm{GAE},c}^{\pi_{\theta_k}}$ is the cost advantage estimate, $\tilde{\gamma}$ is an effective discount factor, and $\beta_q > 0$ controls the constraint regularization.

CUP can thus be interpreted as a first-order projection method that enforces constraint satisfaction by explicitly projecting policy updates onto a linearized feasible set, while maintaining stability through KL regularization.

### J.5.6. P3O (ZHANG ET AL., 2022)

Penalized Proximal Policy Optimization (P3O) formulates constrained policy optimization as an unconstrained problem by introducing an exact penalty constructed via hinge (ReLU) operators.

Given a CMDP with constraints $J_{c_i}(\pi) \leq d_i$, P3O optimizes the following penalized objective:

$$
\begin{aligned}
\theta_{k+1} = \arg\min_{\theta}\ & \mathbb{E}_{s\sim d^{\pi_k},\, a\sim\pi_\theta}[-A_R^{\pi_k}(s,a)] \\
& + \kappa \sum_i \max\Big\{0,\ \mathbb{E}_{s\sim d^{\pi_k},\, a\sim\pi_\theta}\big[A_{C_i}^{\pi_k}(s,a)\big] + (1-\gamma)\big(J_{C_i}(\pi_k) - d_i\big)\Big\},
\end{aligned}
\tag{67}
$$

where $\kappa > 0$ is a penalty factor, $A_R^{\pi_k}$ and $A_{C_i}^{\pi_k}$ denote reward and cost advantages, and the hinge operator activates only when the corresponding constraint is violated.

The penalty term vanishes when all constraints are satisfied, reducing (67) to standard policy optimization in the feasible region. Crucially, P3O interprets the hinge penalty as an *exact penalty function*. Specifically, it is shown that if $\bar{\lambda}$ denotes the optimal Lagrange multiplier vector of the original constrained problem, then for any $\kappa \geq \|\bar{\lambda}\|_\infty$, the penalized problem (67) shares the same set of optimal solutions as the original constrained optimization problem.

### J.5.7. IPO (LIU ET AL., 2020)

Interior-point Policy Optimization (IPO) addresses constrained policy optimization by incorporating barrier functions directly into the objective, following the classical interior-point theory.

Given constraints $J_{C_i}(\pi) \leq d_i$, IPO introduces the shifted constraint

$$
\bar{g}_i(\pi) := J_{C_i}(\pi) - d_i,
$$

and replaces the hard feasibility requirement $\bar{g}_i(\pi) \leq 0$ with a logarithmic barrier. The resulting unconstrained objective takes the form

$$
\max_{\theta}\ \mathcal{L}_{\mathrm{IPO}}(\theta) = \mathcal{L}_{\mathrm{CLIP}}(\theta) + \sum_i \phi\big(\bar{g}_i(\pi_\theta)\big),
\tag{68}
$$

where $\mathcal{L}_{\mathrm{CLIP}}$ denotes the PPO clipped surrogate objective and

$$
\phi(x) = \frac{1}{t}\log(-x), \qquad x < 0,
\tag{69}
$$

is a logarithmic barrier function with temperature parameter $t > 0$.

The barrier term diverges to $-\infty$ as $\bar{g}_i(\pi_\theta) \uparrow 0$, preventing iterates from crossing the constraint boundary. Increasing $t$ improves the approximation to the hard feasibility indicator but simultaneously induces steeper gradients near the boundary.

### J.5.8. EPO (GAO ET AL., 2024)

Exterior Penalty Policy Optimization (EPO) formulates constrained policy optimization as an unconstrained problem using an exterior penalty function. Starting from the surrogate constrained objective, EPO defines the reward and constraint surrogates

$$
F_R(\pi) = \mathbb{E}_{s\sim d_{\pi_k},\, a\sim\pi}[A_R^{\pi_k}(s,a)], \qquad F_C(\pi) = J_C(\pi_k) + \frac{1}{1-\gamma}\mathbb{E}_{s\sim d_{\pi_k},\, a\sim\pi}[A_C^{\pi_k}(s,a)] - d.
$$

Constraint violations are penalized through a learned penalty metric

$$
\Phi(F_C^+(\pi)) = \alpha F_C^+(\pi) + (1-\alpha)\big(F_C^+(\pi)\big)^2, \qquad F_C^+(\pi) = \max\{F_C(\pi), 0\},
$$

where $\alpha \in [0, 1]$ balances linear and quadratic penalties. The resulting exterior penalty objective is

$$\max_{\pi \in \Pi_\theta} \; F_R(\pi) - \frac{1}{\mu} \, \Phi\big(F_C^+(\pi)\big), \tag{70}$$

with penalty factor $\mu > 0$.

EPO employs a *Penalty Metric Network* to adapt $\Phi(\cdot)$ across near- and far-boundary regions of the constraint space, enabling stronger penalties for large violations and smoother corrections near feasibility. By solving a sequence of penalty problems with decreasing $\mu_t \to 0$, EPO guarantees monotonic reduction of constraint violations and convergence to a feasible optimum under suitable conditions.

### J.5.9. APPO (DAI ET AL., 2023)

Augmented Proximal Policy Optimization (APPO) follows exactly the Augmented Lagrangian methods described in J.3. Using the local surrogate reward $L_R^{\pi_k}(\pi) = \mathbb{E}_{s \sim d_{\pi_k}, a \sim \pi}\big[A_R^{\pi_k}(s, a)\big]$ and constraint surrogate $\phi_{c_i}^{\pi_k}(\pi) = J_{c_i}(\pi_k) + \frac{1}{1-\gamma}\mathbb{E}_{s \sim d_{\pi_k}, a \sim \pi}\big[A_{c_i}^{\pi_k}(s, a)\big] - b_i$, APPO considers the equality-constrained reformulation

$$\phi_{c_i}^{\pi_k}(\pi) + p_i = 0, \qquad p_i \geq 0,$$

and constructs the augmented Lagrangian

$$\mathcal{L}(\pi, \lambda, \sigma) = -L_R^{\pi_k}(\pi) + \sum_i \left[ \frac{\sigma}{2}\Big( \max\Big\{ \frac{\lambda_i}{\sigma} + \phi_{c_i}^{\pi_k}(\pi), 0 \Big\}^2 - \frac{\lambda_i^2}{\sigma^2} \Big) \right], \tag{71}$$

where $\lambda_i \geq 0$ are Lagrange multipliers and $\sigma > 0$ is a penalty factor.

The resulting primal gradient takes the form

$$\nabla_\pi \mathcal{L} = -\nabla_\pi L_R^{\pi_k}(\pi) + \sum_i \mathbf{1}\big\{ \phi_{c_i}^{\pi_k}(\pi) \geq -\tfrac{\lambda_i}{\sigma} \big\} \big( \lambda_i + \sigma \phi_{c_i}^{\pi_k}(\pi) \big) \nabla_\pi \phi_{c_i}^{\pi_k}(\pi), \tag{72}$$

reducing oscillations and dual-lag with the early activation of the penalty when the constraint violation exceeds the threshold $-\frac{\lambda}{\sigma}$. APPO updates the multipliers via

$$\lambda_{i,k+1} = \big[ \lambda_{i,k} + \sigma \, \phi_{c_i}^{\pi_k}(\pi_k) \big]_+. \tag{73}$$

APPO establishes that, for sufficiently large but finite $\sigma$, the augmented problem is *exact*, i.e., it shares the same optimal solution set as the original constrained problem. This avoids the infinite or very large penalty factors required by classical penalty methods, while stabilizing constraint satisfaction through the quadratic deviation term.

### J.5.10. CPPOPID (STOOKE ET AL., 2020)

Constraint-Controlled PPO with PID Lagrangian (CPPOPID) extends the classical primal–dual approach by modifying the *dual update dynamics* rather than the primal objective. Starting from the standard Lagrangian formulation with constraint $g(\pi) = J_C(\pi) - d \leq 0$, CPPOPID retains the Lagrangian objective $\mathcal{L}(\pi, \lambda) = J(\pi) - \lambda g(\pi)$, but replaces the integral-only dual ascent update with a proportional–integral–derivative (PID) controller.

To stabilize learning when $\lambda$ becomes large, CPPOPID introduces a rescaled primal objective

$$\theta^*(\lambda) = \arg\max_\theta \frac{1}{1+\lambda}\big( J(\pi_\theta) - \lambda J_C(\pi_\theta) \big), \tag{74}$$

which preserves the direction of the Lagrangian gradient while normalizing its magnitude. Equivalently, defining $u = \frac{\lambda}{1+\lambda} \in [0, 1]$, the policy gradient can be written as

$$\nabla_\theta \mathcal{L} = (1 - u) \nabla_\theta J(\pi_\theta) - u \nabla_\theta J_C(\pi_\theta). \tag{75}$$

The Lagrange multiplier is updated using a PID control rule,

$$\lambda_k = \big( K_P \Delta_k + K_I \sum_{t \leq k} \Delta_t + K_D (\Delta_k - \Delta_{k-1})_+ \big)_+, \qquad \Delta_k = J_C(\pi_k) - d, \tag{76}$$

where $K_P, K_I, K_D \geq 0$ are proportional, integral, and derivative gains.

CPPOPID can thus be viewed as a *primal–dual method with controlled dual dynamics*: the primal update remains Lagrangian, while proportional and derivative terms in the multiplier update aim to reduce oscillations and overshoot caused by integral-only dual ascent but in return introducing more hyperparameters to tune.

