# OpenReview forum: "CSPO: Constraint-Sensitive Policy Optimization for Safe Reinforcement Learning"
_ICML.cc/2026/Conference — ICML 2026 spotlight_

### Official Review · Reviewer_YYAt · 2026-02-16

**Soundness:** 3
**Presentation:** 3
**Significance:** 3
**Originality:** 3
**Overall Recommendation:** 5
**Confidence:** 4

**Summary:**

This paper studies constrained RL and introduces a gradient-sensitive correction term into a first-order primal–dual framework to mitigate overshooting and oscillations during training. The key idea is to update conservatively near steep constraint boundaries and more aggressively in flatter regions. Theoretical results show (1) it preserves the original KKT solution set, and (2) the method enjoys standard inner-loop stationarity guarantees. Empirically, the paper evaluates performance using average return/cost and three training-dynamics metrics—violation frequency, time-to-safety, and reward preservation—showing improved safety recovery and reduced oscillations compared to primal–dual and penalty baselines.

**Compliance With Llm Reviewing Policy:**

Affirmed.

**Final Justification:**

Overall, I find the paper well-motivated, with promising empirical results and a reasonable theoretical foundation. The rebuttal addressed my main concerns, particularly clarifying the bridge between the geometric intuition and the final objective, strengthening the discussion around $\alpha$, and adding useful empirical diagnostics. While I would like to see these clarifications cleanly integrated into the final version, my evaluation remains positive.

**Key Questions For Authors:**

- The paper claims faster safety recovery. Intuitively this relates to convergence speed. Could you discuss their connection, and is it possible to provide theoretical insight about convergence rates (e.g., versus naive PPO-Lagrangian / lagging-dual baselines) rather than only final convergence properties (Section 3.3)? Also, Figure 3 does not seem to show a clear advantage in convergence speed.
- In Table 1, CSPO’s cost in Hopper and Swimmer is far below the constraint threshold, which may indicate over-conservatism (despite strong reward). Can the authors explain why this happens and whether it is desirable?
- Section 5.4 mentions a violation-adaptive schedule, but its motivation and design choices are not sufficiently clear. Could you explain why the schedule is designed in this specific form?
- Would it be helpful to report worst-case safety behavior during training (e.g., the maximum violation magnitude per episode)? This could provide an additional view of safety recovery and peak overshoot during oscillatory phases
- In Section 3.2, $\eta_\lambda$ is not defined when first introduce.
- In Figure 1, adding arrows to indicate the policy update direction along the curves may improve interpretability.

**Limitations:**

Yes

**Strengths And Weaknesses:**

## Strengths

- **Presentation:** Overall well-written and easy to follow. The method and experiments are clearly described and structured.
- **Soundness:** It provides theoretical analysis and, at a high level, the results seem consistent with the stated assumptions (I did not identify obvious issues, though I did not verify every proof line-by-line). The empirical evaluation is promising, covering a reasonably broad set of environments and strong baselines. The proposed training-dynamics metrics are appropriate and help diagnose oscillation and safety recovery behavior.
- **Significance:** The oscillation/overshooting issue in constrained RL is real and practically relevant. The gradient-sensitive correction intuition is reasonable and potentially useful for stabilizing training.

## Weaknesses

- **Presentation:** The paper motivates the approach from a gradient-step argument (Eq. 11), but the final method is introduced as an additional loss term (Eq. 12). The bridge between these two perspectives could be further clarified and explained.
- **Significance:** The approach requires task-specific tuning. Table 4 suggests the best $\alpha$ differs substantially across environments, and Figure 4 further indicates sensitivity. More empirical guidance for selecting $\alpha$ could strengthen the practical impact.
- **Originality:** While the gradient-sensitive correction is a sensible addition, the novelty at the algorithmic level appears a bit modest since the method seems to be one incremental trick on top of existing primal–dual framework.

---

> ### Author Rebuttal · Authors · 2026-03-30
>
> We thank the reviewer for the feedback
>
> **W1. Presentation: Bridge between Eq (11) and Eq (12)**
> We clarify that Eq. (11) is the magnitude of the smallest one-step parameter update $\Delta\theta^\star$ that reduces violation by a fraction $\alpha$. $q_k(\theta) = \frac{\alpha}{2} w_k [g(\theta)]+^{2}$ (Eq. 12) is a smooth differentiable function of $\theta$ whose gradient $\nabla q_k(\theta) = \alpha\frac{[g(\theta)]+}{\|\nabla g(\theta_k)\|^2} \nabla g(\theta)$ recovers $\Delta\theta^\star$ at $\theta=\theta_k$, turning the geometric update into a differentiable objective optimizable with standard gradient methods. Substituting into the Lagrangian gradient yields the effective multiplier $\lambda_{\text{eff}} = \lambda + \alpha w_k [g(\theta)]+$. We will better clarify this connection in the revised version
>
>
> **W2. Task-specific tuning of $\alpha$**
> We refer the reviewer to our detailed response to Reviewer tNSa (W2), which provides empirical guidance for selecting $\alpha$ based on the cost structure of the target environment
>
> **W3. Originality**
> The novelty of CSPO lies in how constraint sensitivity is derived and integrated into the optimization dynamics. $w_k = \|\nabla g(\theta_k)\|^{-2}$ is not a tunable scaling factor, it arises from the shortest-distance formulation (Section 3.1) as the principled correction strength. Existing primal-dual methods apply uniform corrections regardless of local constraint sensitivity, causing slow recovery in flat regions or overshoot near steep boundaries. In contrast, CSPO induces an effective multiplier $\lambda_{\text{eff}} = \lambda + \alpha w_k [g(\theta)]_+$ that adapts to local constraint geometry
>
> This changes how violations are corrected during optimization. The improvements in Table 1 and Fig. 2-3 reflect this qualitative difference and are not easily reproduced by adjusting learning rates or penalty coefficients in existing methods
>
> **Q1. Convergence speed vs. safety recovery speed**
> We distinguish safety recovery speed (transient return to feasibility) from convergence speed (asymptotic rate to a stationary point). While related, CSPO specifically targets the former to mitigate dual-lag and oscillations. Prop. 4.2 provides a local decrease condition on constraint violation under $g(\theta)>0$, introducing a state-dependent correction that strengthens updates when violations persist; this is consistent with the faster Time-To-Safety (TTS) observed in Fig. 2
>
> Our analysis focuses on inner-loop stationarity (Prop. 4.1) and does not provide explicit global convergence rate comparisons. Establishing such rates for non-convex, stochastic deep RL settings for CSPO, remains an important direction for future work.
>
> In Fig. 3, CSPO’s advantage is not a steeper reward curve, but reduced cost oscillations and earlier stabilization within the feasible region compared to Lagrangian baselines
>
> **Q2. Hopper and Swimmer cost well below threshold.**
> In Hopper and Swimmer, we observed a relatively flat constraint surface during early training (small $\nabla g$ in the first ~8 epochs), letting CSPO push the policy aggressively toward the feasible region (large $w_k$). Once it becomes feasible, the correction term deactivate, the optimization becomes a pure reward maximization. Specifically in these 2 environments, it is relatively easier to reach high reward without pushing the velocity constraint near the boundary, making the average cost stay well below the threshold. The observed low cost is thus a combination of CSPO's exploitation of the local constraint geometry and a task structure that permits high reward without frequent constraint boundary interaction
>
> **Q3. Adaptive $\alpha$ schedule motivation.**
> Motivated by Proposition 4.2 and Remark 4.3: the constraint decrease threshold scales as $1/\alpha$, so larger violations should trigger more aggressive correction. Setting $\alpha_k = \text{clip}([J_c(\pi_k) - d]+/(d+\epsilon), 0, 1)$ (where $[J_c(\pi_k) - d]+$ is the current violation) directly encodes this, $\alpha_k$ is small near feasibility and $\alpha_k \approx 1$ under large violations, making $\alpha$ scale with violation magnitude. Normalization by $d$ makes the schedule budget-invariant
>
> **Q4. Worst-case violation magnitude.**
> We agree this is a useful metric and we computed it for the rebuttal (table below) and will be added to Table 5. CSPO achieves lower cost peaks during training compared to baselines, confirming that CSPO's sensitivity-aware correction reduces peak overshoot during oscillatory phases, consistent with the smoother cost curves in Figure 3
>
> | Env | CSPO | PPO-Lag | APPO | CPPOPID |
> |---|---|---|---|---|
> | Ant | **11.3±3.4** | 55.5±10.0 | 12.6±1.8 | 28.2±4.2 |
> | PointGoal | **25.7±31.6** | 69.0±12.5 | 73.8±41.0 | 55.7±28.7 |
>
>
> **Q5,Q6. $\eta_\lambda$ and Figure 1 arrows**
> $\eta_\lambda$ is the dual step size (learning rate) and will be clearly defined in Section 3.2. Directional arrows will be added to Figure 1 in the camera-ready version

---

> > ### Author Rebuttal · Reviewer_YYAt · 2026-04-02
> >
> > Thank the authors for the clarifications and additional experiments—most of my concerns are addressed. One minor suggestion: since performance is sensitive to $\alpha$, and the adaptive schedule helps mitigate this, and the authors’ claim that an adaptive schedule can mitigate this issue, it may be worth highlighting this mechanism more prominently in future versions, possibly even mentioning it in the methods section.

---

> > > ### Author Response · Authors · 2026-04-02
> > >
> > > We thank the reviewer for the positive engagement and will highlight the adaptive schedule more prominently in the final version.

---

### Official Review · Reviewer_tNSa · 2026-03-02

**Soundness:** 3
**Presentation:** 3
**Significance:** 2
**Originality:** 2
**Overall Recommendation:** 5
**Confidence:** 3

**Summary:**

This paper proposes Constraint-Sensitive Policy Optimization (CSPO), a first-order primal–dual method that integrates local constraint sensitivity into the policy update process. CSPO enhances the primal objective by introducing a correction term based on the shortest geometric distance to the constraint boundary. CSPO augments the primal update with a constraint correction that is applied only when violations occur. To scale this correction, the norm of the constraint gradient is used. Even though the objective is modified, the original optimal solution is preserved. The method is analyzed theoretically, and it is empirically shown that they lead to fewer constraint violations.

**Compliance With Llm Reviewing Policy:**

Affirmed.

**Final Justification:**

The authors’ clarification satisfactorily addresses my concerns. Furthermore, the proposed method consistently achieves higher rewards across most tasks compared to the baseline approaches. In light of this, I am inclined to increase my score.

**Key Questions For Authors:**

What is the novelty of this work beyond combining the shortest geometric distance to the constraint boundary?
See strengths and weaknesses.

**Limitations:**

yes

**Strengths And Weaknesses:**

Strength:
1. The paper tackles an important problem in safe reinforcement learning using first-order primal–dual approaches.
2. The paper is clearly written, theoretically sound, and logically organized.
3. The experimental analysis is comprehensive.

Weakness:
1. The author used sensitivity-based normalization that could be unreliable in sparse settings where constraint gradients are noisy.
2. CSPO introduces an additional hyperparameter α, which controls the strength of constraint correction. Its optimal value is task-dependent and requires manual tuning.
3. While the author states that the additional computational overhead is minimal, it would be helpful to include quantitative runtime comparisons or wall-clock training costs to substantiate this claim.
4. While the proposed method achieves strong reward performance and maintains the cost within the specified limits, its ability to reduce the cost is relatively limited.
5. In most tasks (see Table 1), the proposed method achieves good rewards, but the cost performance is weaker compared to existing methods.

---

> ### Author Rebuttal · Authors · 2026-03-30
>
> We thank the reviewer for the positive assessment of clarity, soundness, and experimental coverage
>
> **W1. Sensitivity normalization in sparse settings.**
> We agree this is a valid limitation and acknowledged it in Section 5.5. The 3 stabilization measures in Appendix F ($\epsilon$-regularization, clipping, EMA smoothing) act as a safeguard to mitigate and reduce noise in sparse or discontinuous settings. In our experiments, these reduce $w_k$ variance by ~2% across all 9 environments, already yielding stable estimates in our reward-dense setting, suggesting that it would provide additional robustness against noisy gradients in sparse environments. We will add this observation about the empirical variance reduction in Appendix F
>
> **W2. Additional hyperparameter $\alpha$.**
> We clarify that $\alpha$ has a direct geometric interpretation as the fractional violation reduction factor (Section 3.1), making its effect predictable and interpretable. In practice, only two values are used across all 9 tasks (Table 4), suggesting per task-family tuning suffices. Moreover, the adaptive schedule in Section 5.2 and Appendix G removes manual tuning and achieves comparable performance.
>
> Our empirical results, together with the cost definitions in Safety Gymnasium, provide practical guidance for this choice: in locomotion tasks, violations often occur as sudden, large increases in cost (e.g., largely exceeding a velocity limit), whereas in navigation tasks they tend to accumulate more gradually and remain bounded by the environment layout. Larger $\alpha$ therefore promotes faster recovery in locomotion while smaller $\alpha$ suffices for navigation. The specific values $\alpha=0.85$ (locomotion) and $\alpha=0.3$ (navigation) were selected via hyperparameter search on one representative environment per family, then fixed across all tasks. More generally, $\alpha$ can be set based on the expected severity of constraint violations, and the adaptive schedule helps when this is unknown a priori
>
> **W3. Runtime overhead.**
> CSPO adds one extra gradient computation per episodic iteration to compute $w_k$. Measured across 5 seeds on PointGoal (on the same hardware and training parameters), total training time is 515 min for CSPO vs 507 min for PPO-Lag, an overhead of **1.3%** over the full runs. We will add this result to the Computational complexity discussion in the paper
>
> **W4–W5. Cost performance.**
> In the Constrained MDP framework, the objective is to maximize reward $J_r(\pi)$ subject to the cost constraint $J_c(\pi)\leq d$, which differs from maximizing the reward *and* minimizing the cost (multi-objective setting). When the *unconstrained* optimum is infeasible, the *constrained* optimum is typically active, i.e., $J_c(\pi^*) \approx d$ (KKT complementary slackness) meaning that the optimal policy lies at the constraint boundary and any unused safety budget therefore corresponds to potential reward that could be achieved while remaining feasible.
>
> From this perspective, methods that maintain cost significantly below the threshold may be overly conservative, potentially sacrificing reward (for example EPO in Table 1). In contrast, CSPO tends to operate closer to the constraint boundary, effectively using the available safety budget, which explains its higher constrained returns in Table 1. As long as $J_c(\pi)\leq d$, the safety requirement is satisfied; thus $J_c \approx d$ reflects efficient safety budget usage rather than limited cost reduction.
>
> From a practitioner's standpoint, if the safety budget was set to $d=25$, a method that delivers a cost of $5$ but half the reward is underperforming compared to CSPO, which delivers the full reward at a cost of $24.5$
>
> **Q1. novelty beyond geometric distance.**
> The contribution is not the shortest-distance derivation itself, but how it is used to *modulate the optimization dynamics*. Specifically, incorporating the shortest-distance into the optimization objective makes CSPO navigate its way back to the feasible region more intelligently by adapting the correction strength via $w_k = \|\nabla g(\theta_k)\|^{-2}$, yielding an effective multiplier $\lambda_{\text{eff}} = \lambda + \alpha w_k [g(\theta)]_+$. This induces *state-dependent correction*: stronger updates in flat regions (where violations persist) and weaker updates near steep boundaries (to avoid overshooting). Existing primal–dual and penalty methods apply uniform corrections and do not account for this local sensitivity and thus tend to overshoot and oscillate as illustrated in Figure 1.

---

> > ### Author Rebuttal · Reviewer_tNSa · 2026-04-01
> >
> > The authors have clarified all the points. In my opinion, the idea of the paper is promising, and the rewards achieved by CSPO are strong across many tasks. However, the cost remains higher than that of other baselines. In safety-critical applications, minimizing cost (i.e., constraint violations) is of primary importance, and even small increases in violations can be undesirable. Therefore, despite the strong reward performance, the safety trade-off compared to existing methods remains a concern.
> > For this reason, although I appreciate the rebuttal, my overall assessment remains unchanged, and I would like to maintain my current score.

---

> > > ### Author Response · Authors · 2026-04-02
> > >
> > > We thank the reviewer for the thoughtful engagement. We respectfully further address the safety concern raised:
> > >
> > > As quoted by [Ray et al. (2019, Section 3.5)](https://cdn.openai.com/safexp-short.pdf) in **Benchmarking Safe Exploration in Deep Reinforcement Learning** (a key  paper in the safe RL literature, published by OpenAI ): *"there is generally a saturation point where the safety requirement is satisfied, and further decreasing the value of the function no longer makes the system meaningfully or usefully safer."* In line with this, once the safety requirement is met, pursuing further cost reduction is not necessarily desirable, as it may push the method toward overly conservative behavior. CSPO is designed precisely around this principle, aiming to satisfy the safety constraint while avoiding unnecessary conservatism that could reduce reward.
> > >
> > > That said, we also empirically assess how CSPO ranks in terms of cost minimization relative to the baselines (see table below), using the results reported in Table 1. CSPO ranks **1st** or **2nd** in **4/9** environments and remains among the top **4** methods in all tasks (except *Ant*), demonstrating highly competitive cost reduction while maintaining a clear advantage in reward.
> > >
> > > | Environment | CSPO Cost | CSPO Rank (↑ = worse) |
> > > |---|---:|---:|
> > > | CarButton | 23.4 | **1/11** |
> > > | CarGoal | 24.0 | **1/11** |
> > > | Hopper | 7.1 | **1/11** |
> > > | Swimmer | 4.8 | 2/11 |
> > > | PointGoal | 23.7 | 2/11 |
> > > | HalfCheetah | 15.5 | 3/11 |
> > > | PointButton | 24.4 | 4/11 |
> > > | Humanoid | 18.0 | 4/11 |
> > > | Ant | 24.5 | 7/11 |
> > >
> > > For applications with stricter safety requirements, one can simply **increase `α`** to induce more conservative behavior. Since `α` has an interpretable effect as a constraint reduction factor (see Section 3.1, Proposition 4.2), also illustrated in Figure 4, practitioners can explicitly control the reward-safety tradeoff without modifying the algorithm.
> > >
> > > We hope this clarifies the reviewer’s concern. Overall, CSPO does not sacrifice safety: it achieves high reward while remaining one of the most constraint-respecting and cost-minimizing methods in the comparison, as shown empirically.

---

### Official Review · Reviewer_Rtu4 · 2026-03-11

**Soundness:** 3
**Presentation:** 3
**Significance:** 3
**Originality:** 2
**Overall Recommendation:** 5
**Confidence:** 4

**Summary:**

This work proposes a new method for Constrained Reinforcement Learning that augments oscillatory primal-dual optimization with a constraint correction term scaled by the cost gradient’s norm.

**Compliance With Llm Reviewing Policy:**

Affirmed.

**Final Justification:**

During the rebuttal the authors have addressed my concerns and I hence updated my pre-rebuttal score of 4 to 5.

**Key Questions For Authors:**

For the results in table 1: Did you use the same alpha or a different one for every problem?

**Limitations:**

Yes

**Strengths And Weaknesses:**

The paper is well written and proposes a simple and seemingly effective way to improve the traditional Lagrange updates. The additional metrics (RP, VF, and TTS) seem meaningful.

One weakness of this method is the strong reliance on accurate advantage estimation: looking at $\nabla g(\theta)$, we have a direct correspondence with $A_c^{\pi_{\theta_k}}$, so I would expect any bias or noise in that estimate to directly affect the $w_k$ term. I’m not sure how big of an issue this is (e.g. CPO has the same issue) but the “dual use” of $g(\theta)$ might increase the errors here.

The constraint sensitivity estimate also seems somewhat oversimplified: Usually, the sensitivity would be estimated using the hessian, e.g. using a memory efficient power iteration
$$v_{k+1} = \frac{Hv_k}{\|Hv_k\|}$$
This only needs Hessian vector products and should be straightforward and memory efficient to compute. An alternative could be Hutchinson trace estimation since $z^T H z \leq \lambda_{\max}\|z\|^2$, or Stochastic Lanczos Quadrature.

While this is a bit more expensive, right now I have a feeling that the $w_k$ estimation would break for eccentric hessians.

While the experiments are generally sufficient, I’m not sure whether 5 experiments are sufficient for making statements: At the very least, I would expect a robust estimator of the mean (i.e. median, trimmed mean, etc…) to prevent outliers from affecting the results.

Finally, I would like to see a more thorough comparison against the PID based approaches. Those (intuitively) seem to follow a similar rational where one tries to reduce the oscillation of multipliers using the derivative term: Arguably, you could frame your method as a variant of the PID based ones with automatic tuning of the “D” term.

---

> ### Author Rebuttal · Authors · 2026-03-30
>
> We thank the reviewer for the feedback.
>
> **W1. Expect any bias / noise to affect w_k**
> We acknowledge this effect in the paper, and mitigate it with: 1) by detaching and treating $w_k$ as a fixed scalar during inner-loop updates, making noise influence the *magnitude* rather than the *direction* of the correction. 2) The stabilization measures in Appendix F ($\epsilon$-regularization, clipping, and EMA smoothing) act as safeguards against noisy estimates, analysis on our experiments show a variance reduction of 2% across tasks, suggesting that they would provide additional robustness in sparser or noisier settings, this observation will be added to Appendix F
>
> **W2. Constraint sensitivity oversimplified**
> We agree that curvature-aware estimates are a reasonable alternative, especially in the case of *eccentric* Hessians. That said, CSPO is a *first-order* method whose correction is derived from the shortest distance to the constraint boundary. Under trust-region bounded updates $\|\Delta\theta\| \leq \epsilon$, the change in $g$ is dominated by its first-order term $O(\epsilon)$, while the second order remainder is bounded by $\frac{1}{2}\lambda_{\max}(H_g)\epsilon^2$, making $\|\nabla g\|^2$ the relevant sensitivity signal in the small-step regime enforced by PPO clipping and early KL stopping (Line 12, Algorithm 1)
>
> Based on the reviewer's suggestion and to test Hessian-based sensitivity in practice, we implemented CSPO-H, using power iteration to estimate $\lambda_{\max}$ as the sensitivity measure instead of $\|\nabla g\|^2$. CSPO-H performs worse and violates safety constraints more frequently (table below). Training showed that $\lambda_{\max}$ is consistently larger than $\|\nabla g\|^2$ ($5.2$x). By Proposition 4.2, CSPO-H's sufficient decrease condition becomes $g(\theta_t) > \delta\lambda_{\max}/(\alpha\|\nabla g\|^2)$ vs. $g(\theta_t) \gtrsim \delta/\alpha$ for CSPO, making the activation threshold larger, thus resulting in slower constraint contraction
>
> Constraint-violating runs are marked with *
>
> | Env | CSPO (R/C) | CSPO-H (R/C) |
> |---|---|---|
> | Ant | **3223.5±71.5** / 24.5±0.6 | 3150.1±68.1 / 22.3±0.4 |
> | Humanoid | **6474.3±64.4** / 18.0±3.1 | 6514.5±75.7 / 26.3±4.0* |
> | PointGoal | **23.8±0.6** / 23.7±0.4 | 22.6±0.3 / 22.6±0.6 |
> | PointButton | **10.1±0.2** / 24.5±0.6 | 10.7±2.0 / 26.6±0.2* |
> | CarGoal | **27.1±0.8** / 24.0±1.0 | 25.27±0.6 / 25.2±0.7* |
>
>
> **W3. Statistical robustness.**
> We computed for the rebuttal, the IQM (mean over interquartile range) for Table 1. CSPO still leads in 8/9 environments with almost no drop in performance. HalfCheetah is the only exception where IQM reveals small sensitivity to outliers:
>
> |Env | CSPO | Best Baseline |
> |---|---|---|
> |Ant|**3231.6±84.8**|PPO-Lag:3134.2±92.0|
> |Humanoid|**6496.8±74.3**|CPO:6213.1±126.5|
> |HalfCheetah|2007.1±415.9|FOCOPS:**2870.7±73.7**|
> |Hopper|**1691.9±7.0**|CUP:1661.6±11.9|
> |Swimmer|**91.4±56.3**|APPO:74.7±55.9|
> |PointGoal|**23.7±0.7**|APPO:23.1±0.3|
> |PointButton|**10.1±0.2**|PPO-Lag:9.1±1.6|
> |CarGoal|**27.1±0.9**|APPO:26.0±1.0|
> |CarButton|**0.7±0.3**|APPO:0.3±0.1|
>
> We note that 5 seeds is standard practice in safe RL papers (e.g., APPO, C-TRPO, EPO, PPO-Lag, CPPOPID)
>
> **W4. Comparison to PID variants.**
> We first clarify that CSPO does not modify the Lagrange multiplier update, which follows standard projected ascent $\lambda_{k+1} = [\lambda_{k} + \eta_{\lambda} g(\theta_k)]+$. CSPO's correction $\alpha w_k [g(\theta)]+$ enters the *primal objective*, the effective multiplier $\lambda_{\text{eff}} = \lambda + \alpha w_k [g(\theta)]+$ is an emergent property of the primal gradient (Eq 16), not a modification to how $\lambda$ is updated
>
> While both CSPO and CPPOPID aim to reduce oscillations, CPPOPID's derivative term $K_D(\Delta_k - \Delta_{k-1})$ is a *temporal signal* on the dual update, CSPO's is a *geometric signal* on the primal objective. As a result, CSPO is independent of the specific dual update rule (e.g., gradient ascent or PID control). Empirically, CSPO outperforms CPPOPID in 8/9 environments (Table 1) with consistently lower TTS (Figure 2) and smoother cost curves (Figure 3). We also added for the rebuttal a recent PID variant [APD](https://arxiv.org/abs/2402.00355) where CSPO outperforms it in 7/8 environments (see table below), with the largest gaps in navigation tasks, suggesting constraint-aware primal scaling addresses failure modes that dual-side adaptation alone may not
>
> |Env|CSPO|APD|
> |---|---|---|
> |Ant|**3223.5±71.5**|3063.1±62.6|
> |Humanoid|**6474.3±64.4**|5622.6±112.9|
> |HalfCheetah|2273.9±335.5|**2624.6±392.1**|
> |Hopper|**1692.6±155.7**|1426.4±494.9|
> |PointGoal|**23.8±0.6**|16.5±4.1|
> |PointButton|**10.1±0.2**|1.8±0.5|
> |CarGoal|**27.1±0.8**|18.4±2.6|
> |CarButton|**0.7±0.3**|-0.6±0.3*|
>
> **Q1. same $\alpha$ per problem?**
> We used *0.3* for navigation and *0.85* for locomotion tasks (Table 4), we refer the reviewer to our response to reviewer tNSa (W2) for a detailed discussion on $\alpha$

---

> > ### Author Rebuttal · Reviewer_Rtu4 · 2026-04-04
> >
> > I thank the authors for their response. My questions have been answered.
> > I hence will increase my score to 5.

---

> > > ### Author Response · Authors · 2026-04-04
> > >
> > > We sincerely thank the reviewer for the engagement and for increasing
> > > the score. We are glad the rebuttal addressed all concerns.

---

### Decision · Program_Chairs · 2026-04-30

**Decision:**

Accept (spotlight)

**Comment:**

This paper addresses a core challenge in primal-dual methods for safe reinforcement learning: oscillations arising from delayed constraint correction. The proposed approach leverages first-order local constraint sensitivity to adapt the strength of constraint enforcement during policy updates, enabling more stable recovery toward feasibility. The method is principled, supported by theoretical analysis, and validated through extensive experiments across diverse environments.

While the initial reviews raised concerns regarding robustness, safety-reward trade-offs, and clarity, these issues were carefully discussed and convincingly resolved during the rebuttal phase through additional explanations and empirical evidence. Following discussion, all reviewers expressed agreement that the paper meets the acceptance criteria. Overall, the work represents a solid and meaningful contribution to safe reinforcement learning and merits acceptance.